# Systemic pharmacological suppression of neural activity reverses learning impairment in a mouse model of Fragile X syndrome

**Amin MD Shakhawat\*, Jacqueline G Foltz, Adam B Nance, Jaydev Bhateja, Jennifer L Raymond\***

Department of Neurobiology, Stanford University, Stanford, United States

**\*For correspondence:**
amins@stanford.edu (AMDS);
jenr@stanford.edu (JLR)

**Abstract** The enhancement of associative synaptic plasticity often results in impaired rather than enhanced learning. Previously, we proposed that such learning impairments can result from saturation of the plasticity mechanism (Nguyen-Vu et al., 2017), or, more generally, from a history-dependent change in the threshold for plasticity. This hypothesis was based on experimental results from mice lacking two class I major histocompatibility molecules, MHCI H2-$K^b$ and H2-$D^b$ (MHCI $K^bD^{b-/-}$), which have enhanced associative long-term depression at the parallel fiber-Purkinje cell synapses in the cerebellum (PF-Purkinje cell LTD). Here, we extend this work by testing predictions of the threshold metaplasticity hypothesis in a second mouse line with enhanced PF-Purkinje cell LTD, the *Fmr1* knockout mouse model of Fragile X syndrome (FXS). Mice lacking *Fmr1* gene expression in cerebellar Purkinje cells (L7-*Fmr1* KO) were selectively impaired on two oculomotor learning tasks in which PF-Purkinje cell LTD has been implicated, with no impairment on LTD-independent oculomotor learning tasks. Consistent with the threshold metaplasticity hypothesis, behavioral pre-training designed to reverse LTD at the PF-Purkinje cell synapses eliminated the oculomotor learning deficit in the L7-*Fmr1* KO mice, as previously reported in MHCI $K^bD^{b-/-}$ mice. In addition, diazepam treatment to suppress neural activity and thereby limit the induction of associative LTD during the pre-training period also eliminated the learning deficits in L7-*Fmr1* KO mice. These results support the hypothesis that cerebellar LTD-dependent learning is governed by an experience-dependent sliding threshold for plasticity. An increased threshold for LTD in response to elevated neural activity would tend to oppose firing rate stability, but could serve to stabilize synaptic weights and recently acquired memories. The metaplasticity perspective could inform the development of new clinical approaches for addressing learning impairments in autism and other disorders of the nervous system.

## eLife assessment

This **important** manuscript follows up on previous findings from the same lab supporting the idea that deficits in learning due to enhanced synaptic plasticity are due to saturation effects. **Compelling** evidence is presented that behavioral learning deficits associated with enhanced synaptic plasticity in a transgenic mouse model can be rescued by manipulations designed to reverse the saturation of synaptic plasticity. In particular, the finding that a previously FDA-approved therapeutic can rescue learning could provide new insights for biologists, psychologists, and others studying learning and neurodevelopment.

## Introduction

Since its discovery, long-term synaptic plasticity has been of great interest to neuroscientists as a therapeutic target for brain disorders, especially disorders affecting learning and memory. Scientific and technological advances have provided an array of tools for enhancing synaptic plasticity. In some cases, experimental manipulations that augment plasticity have succeeded in augmenting learning (*Tang et al., 1999*; *van Praag et al., 1999*; *Lee and Silva, 2009*). However, in many cases, manipulations that augment plasticity have impaired learning (*Migaud et al., 1998*; *Uetani et al., 2000*; *Gu et al., 2002*; *Cox et al., 2003*; *Rutten et al., 2008*; *Navakkode et al., 2022*). Surprisingly, there have been few attempts to reconcile these conflicting findings with a mechanistic explanation for why enhancing synaptic plasticity can have opposite effects on learning. Such mechanistic insight about how enhanced synaptic plasticity functions in vivo could facilitate the development of this approach as a viable clinical intervention for learning disorders, recovery from stroke or brain injury, dementia, addiction, and other neurological and psychiatric disorders.

Recently, we proposed a testable hypothesis about what can go wrong with augmented plasticity in vivo, based on experimental and theoretical analysis of learning in mice with enhanced associative synaptic plasticity in the cerebellum (*Nguyen-Vu et al., 2017*). Associative LTD at the cerebellar PF-Purkinje cell synapses (PF-Purkinje cell LTD) has been implicated in certain cerebellum-dependent learning tasks and not others, based in part on the observation of selective learning impairments in mouse lines with impaired PF-Purkinje cell LTD (reviewed in *Raymond and Medina, 2018*; *De Zeeuw et al., 2021*). Initially, we expected that mice with enhanced PF-Purkinje cell LTD would exhibit the exact opposite behavioral phenotype as mice with impaired PF-Purkinje LTD, that is, enhancement of learning on the same behavioral tasks in which mice with impaired PF-Purkinje cell LTD exhibit impaired learning. Contrary to this expectation, double knockout of the major histocompatibility class I molecules MHCI H2-K$^b$ and H2-D$^b$ (MHCI K$^b$D$^{b-/-}$), which enhances PF-Purkinje cell LTD (*McConnell et al., 2009*), results in the very same, specific oculomotor learning impairment as observed in mice with impaired PF-Purkinje cell LTD (*Nguyen-Vu et al., 2017*). To explain the puzzling observation that the enhancement of a plasticity mechanism could yield the same behavioral phenotype as its impairment, we hypothesized that enhanced LTD prevents learning by allowing spontaneous activity in the circuit to aberrantly recruit and saturate this form of plasticity, making it unavailable at the specific synapses where it is needed to support learning. Two key predictions of this hypothesis were confirmed experimentally by previous work: optogenetic stimulation of the circuit designed to induce PF-Purkinje cell LTD before training recapitulated in WT mice the same, specific oculomotor learning deficit observed in the MHCI K$^b$D$^{b-/-}$ mice with enhanced LTD; and a behavioral manipulation designed to reverse PF-Purkinje cell LTD before oculomotor training reversed the learning deficit in MHCI K$^b$D$^{b-/-}$ mice (*Nguyen-Vu et al., 2017*).

Here, we further tested the hypothesis that the enhancement of PF-Purkinje cell LTD can result in its aberrant recruitment by ongoing neural activity and consequent increased threshold for its induction and reduced availability to support learning. First, we replicated key behavioral findings in a different line of mice with enhanced PF-Purkinje cell LTD. Purkinje cell-specific knock out of the Fragile X gene *Fmr1* enhances PF-Purkinje cell LTD (*Koekkoek et al., 2005*). We show that these L7-*Fmr1* KO mice are selectively impaired on LTD-dependent oculomotor learning tasks, and that their learning deficit can be reversed with behavioral pre-training designed to reverse PF-Purkinje cell LTD, as previously reported in the MHCI K$^b$D$^{b-/-}$ mice with enhanced PF-Purkinje cell LTD. We then test a new prediction about a pharmacological treatment to reverse the learning deficit in mice with enhanced associative synaptic plasticity. The experimental results support and extend the influential concept of a history-dependent sliding threshold for synaptic plasticity (*Bienenstock et al., 1982*) as a regulator of learning.

## Results

### Selective learning impairment in mice with enhanced associative long-term depression in the cerebellum

We assessed oculomotor learning in mice lacking expression of the Fragile X gene *Fmr1* in cerebellar Purkinje cells, which have been shown to have enhanced PF-Purkinje cell LTD in vitro (*Koekkoek et al., 2005*). Purkinje-cell-specific *Fmr1* knock out mice were generated by crossing conditional *Fmr1*

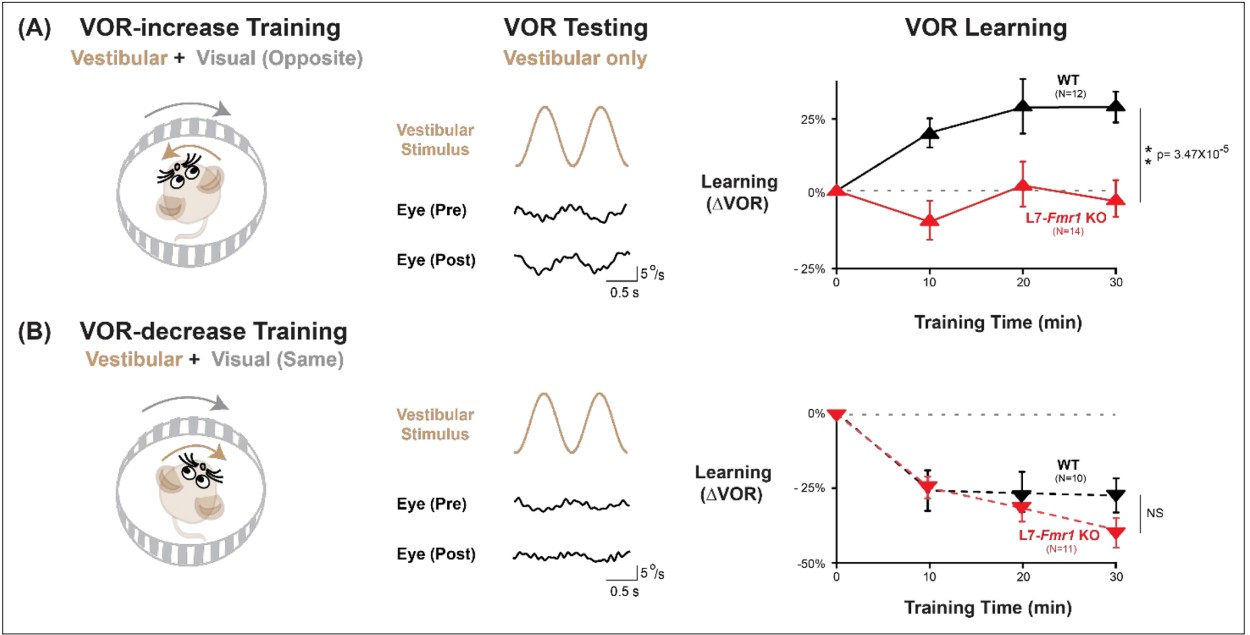

**Figure 1.** VOR-increase learning is impaired in L7-*Fmr1* KO mice with enhanced cerebellar LTD. (**A**) Training to increase the VOR. *Left,* VOR-increase training paired a vestibular stimulus (1 Hz sinusoidal rotation about an earth-vertical axis, *brown*) with oppositely directed visual stimulus motion (*grey*). *Middle,* Example raw eye velocity responses (*black*) to the vestibular stimulus alone in the dark, that is, the VOR, measured Pre and Post VOR-increase training. *Right,* Average learned change in the amplitude of the VOR relative to pre-training (*upward triangles*), measured in the dark after each 10 min VOR-increase training block in the L7-*Fmr1* KO (*red*) and WT mice (*black*). (**B**) Training to decrease the VOR. *Left,* VOR-decrease training paired a vestibular stimulus (1 Hz sinusoidal rotation) with visual stimulus motion in the same direction. *Middle,* Example VOR responses in the dark, measured Pre and Post VOR-decrease training. *Right,* VOR-decrease learning (*downward triangles*). NS = not significant. In this and all figures, values plotted are mean ± SEM.

The online version of this article includes the following source data and figure supplement(s) for figure 1:

**Source data 1.** VOR-increase learning is impaired in L7-*Fmr1* KO mice with enhanced cerebellar LTD.

**Figure supplement 1.** Similar oculomotor learning impairments and efficacy of diazepam pretreatment in male and female L7-*Fmr1* KO mice.

**Figure supplement 2.** Baseline oculomotor performance of L7-*Fmr1* KO mice was indistinguishable from WT.

knockout mice (*Mientjes et al., 2006*) with mice expressing Cre under the control of the L7/Pcp2 promoter (*Zhang et al., 2004*; see Materials and methods). We tested the ability of these L7-*Fmr1* KO mice to adaptively modify their eye movement responses to vestibular and visual stimuli, and compared their performance on different oculomotor learning tasks that have previously been shown to have different sensitivity to perturbations of PF-Purkinje cell LTD.

We first assessed adaptive modification of the vestibulo-ocular reflex (VOR). The VOR stabilizes images on the retina by using the vestibular sensory input caused by a head movement to drive an oppositely directed eye movement response. Learning can adjust the amplitude of this oculomotor reflex to improve the stabilization of visual images on the retina for successful navigation in the world (*Gonshor and Jones, 1973*; *Ito et al., 1974a*; *Miles and Fuller, 1974*; *Broussard and Kassardjian, 2004*; *Gittis and du Lac, 2006*; *Cullen, 2023*). Mice were trained to adaptively increase or decrease their VOR amplitude using two types of vestibular-visual stimulus pairings (*Figure 1*; *Boyden and Raymond, 2003*; *Boyden et al., 2004*). When a vestibular stimulus (1 Hz sinusoidal rotation about an earth-vertical axis with peak velocity of ± 10°/s) was paired with oppositely directed motion of a large-field visual stimulus for 30 min (*Figure 1A*, *left*; see Materials and methods), this induced an adaptive learned increase in the eye movement responses of wild type (WT) mice to the vestibular stimulus alone (VOR-increase learning; *Figure 1A*, *right, black*; p=7.37 × 10⁻⁴, 0 vs 30 min, Tukey). When the vestibular stimulus was instead paired with motion of a visual stimulus in the same direction as the head (*Figure 1B*, *left*), this induced an adaptive learned decrease in the eye movement responses of WT mice to the vestibular stimulus alone (VOR-decrease learning; *Figure 1B*, *right, black*; p=0.001, 0 vs 30 min, Tukey).

Both VOR-increase and VOR-decrease learning are cerebellum dependent (*Ito et al., 1974b*; *Robinson, 1976*; *Lisberger et al., 1984*; *Nagao, 1983*; *Michnovicz and Bennett, 1987*; *Pastor et al., 1994*; *Koekkoek et al., 1997*; *McElligott et al., 1998*). However, manipulations that impair or enhance PF-Purkinje cell LTD have previously been found to selectively alter VOR-increase learning, with less or no effect on VOR-decrease learning (*Li et al., 1995*; *Boyden et al., 2006*; *Hansel et al., 2006*; *Guo et al., 2014*; *Kimpo et al., 2014*; *Nguyen-Vu et al., 2017*; *Kakegawa et al., 2018*; *Zhang et al., 2023*). Accordingly, the L7-*Fmr1* KO mice with enhanced PF-Purkinje cell LTD were selectively and profoundly impaired on VOR-increase learning. Unlike the WT control group, L7-*Fmr1* KO mice exhibited no significant change in the amplitude of their VOR after 30 min of VOR-increase training (*Figure 1A*, *red*; p=0.97, L7-*Fmr1* KO, 0 vs 30 min; p=3.47 × 10⁻⁵, L7-*Fmr1* KO vs. WT, 30 min; Tukey). In contrast, VOR-decrease learning in the L7-*Fmr1* KO mice was robust and indistinguishable from that of their WT littermates (*Figure 1B, red*; p=1.10 × 10⁻⁵, L7-*Fmr1* KO, 0 vs 30 min; p=0.091, L7-*Fmr1* KO vs. WT, 30 min; Tukey). The VOR-increase learning deficit was observed in both male and female L7-*Fmr1* KO mice (*Figure 1—figure supplement 1*). Baseline oculomotor performance of L7-*Fmr1* KO mice was normal, as were the eye movement responses to the paired presentation of visual and vestibular stimuli used for both types of training (*Figure 1—figure supplement 2*), suggesting that there was no deficit in the vestibular, visual or oculomotor functions required to perform the learning tasks; rather the L7-*Fmr1* KO mice have a selective deficit in learning. These results support previous findings that manipulations of PF-Purkinje cell LTD selectively affect VOR-increase learning, and that the enhancement of PF-Purkinje cell LTD impairs rather than enhances this form of learning.

## Behavioral pre-training eliminates learning impairment in L7-*Fmr1* KO mice with enhanced LTD

A key question is why the enhancement of PF-Purkinje cell LTD would impair LTD-dependent learning. One potential explanation is that the enhancement of LTD allows the spontaneous activity in the cerebellar circuit to aberrantly recruit this mechanism and increase the threshold for its further recruitment, reducing its availability to support new LTD-dependent learning. If this is the case, then manipulations that prevent or reverse excessive PF-Purkinje cell LTD before training should reset the circuit to a state compatible with new LTD-dependent learning, and thereby improve VOR-increase learning in the L7-*Fmr1* KO mice. To test this prediction, we first employed a behavioral approach designed to reverse PF-Purkinje cell LTD in the oculomotor cerebellum before training.

In wild-type mice, VOR-decrease training can rapidly reverse any behavioral evidence of prior VOR-increase learning, which suggests that VOR-decrease training can reverse any plasticity induced during VOR-increase learning, including any PF-Purkinje cell LTD (*Boyden and Raymond, 2003*), presumably through the induction of PF-Purkinje cell LTP (*Lev-Ram et al., 2003*; *Shim et al., 2022*). Accordingly, VOR-decrease pre-training was previously found to reverse the oculomotor learning deficit in *MHCI KᵇDᵇ⁻/⁻* mice with enhanced PF-Purkinje cell LTD (*Nguyen-Vu et al., 2017*). We tested whether the same behavioral pre-training intervention could also eliminate the learning deficit in L7-*Fmr1* KO mice (*Figure 2A–C*).

L7-*Fmr1* KO and WT mice were subjected to 30 min of VOR-decrease pre-training followed by 30 min of VOR-increase training. In WT mice, there were adaptive changes in the amplitude of the VOR during both the pre-training and training periods—first a decrease and then an increase in the eye movement response to the vestibular stimulus alone (*Figure 2B*, *black*; VOR-decrease, *dotted lines*, p=0.02, WT –30 vs 0 min; VOR-increase, *solid lines*, p=0.001, WT 0 vs 30 min; Tukey). The L7-*Fmr1* KO mice exhibited adaptive changes in the VOR during both the pre-training and training periods that were statistically indistinguishable from WT (*Figure 2B*, *red*; VOR-decrease, *dotted lines*, p=0.18, L7-*Fmr1* KO vs. WT, 0 min, Tukey; *Figure 2B*, *bar graphs*, p=0.17, L7-*Fmr1* KO vs. WT, VOR-increase from 0 to 30 min, t-test; *Figure 2—figure supplement 1*). Although in the absence of pre-training, VOR-increase training failed to induce any significant change in the VOR of the L7-*Fmr1* KO mice (*Figure 2A*, *red, solid lines and bar graph*; p=0.99, L7-*Fmr1* KO, 0 vs 30 min, Tukey), the same VOR-increase training procedure did induce a significant increase in VOR amplitude when delivered to the same cohort of mice after VOR-decrease pre-training (*Figure 2B*, *red*, *solid lines and bar graph*, p=0.001, 0 vs 30 min, Tukey). In other words, the ability of the L7-*Fmr1* KO mice to learn in response to the VOR-increase training varied with the recent history of experience (*Figure 2*, *compare red bars in A vs. B*; p=0.01, VOR-increase learning of L7-*Fmr1* KO without vs.

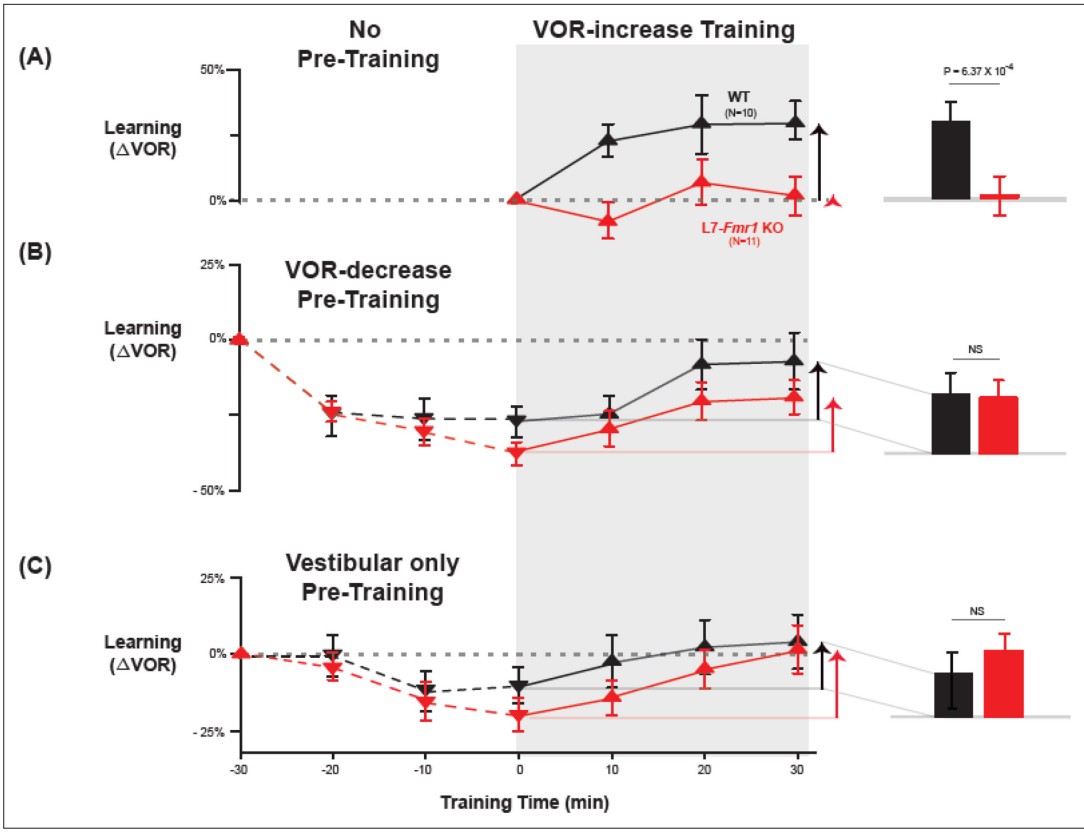

**Figure 2.** Behavioral pre-training rescued learning impairment of L7-*Fmr1* KO mice with enhanced associative LTD. Associative VOR-increase learning (*shaded area* and *bar graphs*), without pre-training (**A**), after VOR-decrease pre-training (**B**), and after Vestibular only pre-training (**C**). (**A**) Learned change in the VOR response measured in the dark after each 10 min block of VOR-increase training in the subset of L7-*Fmr1* KO (*red*) and WT (*black*) mice from *Figure 1A* that were also tested after pre-training. (**B**) Changes in the VOR measured in the dark after each block of VOR-decrease pre-training (*downward triangles, dashed lines*) and then subsequent VOR-increase training (*upward triangles, solid lines*). (**C**) Changes in the VOR measured in the dark after each block of Vestibular only pre-training (*downward triangles, dashed lines*) and then VOR-increase training (*upward triangles, solid lines*). *Right, Arrows* and *bars graphs* show the total change in the VOR induced by 30 min of VOR-increase training (training time = 30) compared with just before VOR-increase training (training time = 0).

The online version of this article includes the following source data and figure supplement(s) for figure 2:

**Source data 1.** Behavioral pre-training rescued learning impairment of L7-*Fmr1* KO mice with enhanced associative LTD.

**Figure supplement 1.** Data from *Figure 2* were subsampled to compare VOR-increase learning in subpopulations of mice matched for the mean learned decrease in the VOR during pre-training.

with pre-training, paired sample t-test). Pre-training experience did not have the same effect in WT mice. The amount of learning exhibited by WT mice in response to VOR-increase training was not enhanced after VOR-decrease pre-training (*Figure 2*, *compare black bars in **A** vs.**B**; p=0.41, paired sample t-test). Thus, VOR-decrease pre-training had different effects on the L7-*Fmr1* KO and WT mice, putting the L7-*Fmr1* KO mice, but not the WT mice, into a state more compatible with VOR-increase learning.

A second behavioral pre-training procedure, habituation of the VOR, induced by presentation of the vestibular stimulus alone in complete darkness (Vestibular only pre-training), had effects similar to those of VOR-decrease pre-training on subsequent VOR-increase learning. After thirty minutes of Vestibular only pre-training, subsequent VOR-increase learning in the L7-*Fmr1* KO mice was comparable to that of their WT littermates (*Figure 2C*, *red* vs. *black* bars; p=0.84, L7-*Fmr1* KO vs. WT, VOR-increase from 0 to 30 min, t-test).

## Pharmacological suppression of neural activity the day before training eliminates learning impairment of L7-*Fmr1* KO mice with enhanced LTD

The preceding results are consistent with the hypothesis (*Nguyen-Vu et al., 2017*) that in mice with enhanced PF-Purkinje cell LTD, spontaneous activity in the circuit can induce LTD and thereby increase the threshold for its subsequent induction, and that behavioral pre-training can alter neural activity in a manner that prevents or reverses this increased threshold for LTD in vivo, thereby reversing the learning impairment. Since PF-Purkinje cell LTD is driven by co-activation of cerebellar parallel fibers and climbing fibers (*Ito and Kano, 1982*; *Ito, 1982*; *Linden and Connor, 1995*), pharmacological suppression of neural activity should also prevent the induction and increase in the threshold for LTD during the pre-training period, and restore the capacity for subsequent LTD-dependent learning in mice with enhanced LTD. We tested this prediction by administering the benzodiazepine diazepam, a positive allosteric modulator of GABA$_A$ receptors, to enhance inhibition and suppress neural activity in the L7-*Fmr1* KO mice during the period preceding VOR-increase training. Diazepam has been shown to reduce neural firing in cerebellar neurons and neural responses to vestibular stimuli (*Ryu and McCabe, 1974*; *Barmack and Pettorossi, 1980*). We assessed VOR learning 2 hr after a single, systemic dose of diazepam, immediately after recovery from diazepam (18–24 hr after administration), and 1 week later.

The acute effect of diazepam administration was to impair learning. There was no effect of diazepam on the baseline amplitude of the VOR response measured in the dark 2 hr after diazepam (*Figure 3—figure supplement 1*), contrary to what has been reported in rabbit (*Barmack and Pettorossi, 1980*). However, when VOR-increase training was delivered 2 hr after systemic administration of diazepam, VOR-increase learning was profoundly impaired in WT as well as L7-*Fmr1* KO mice (*Figure 3—figure supplement 2*).

It is not surprising that the acute effect of suppressing neural activity was to impair learning. The key question was whether this suppression of activity could reset the circuit to a state compatible with subsequent LTD-dependent learning. Therefore, VOR learning was tested after recovery from the acute effects of diazepam. Diazepam has a long half-life of ~24 hr (*Riss et al., 2008*), therefore mice were allowed to recover in their home cage for 18–24 hr after diazepam administration, and then

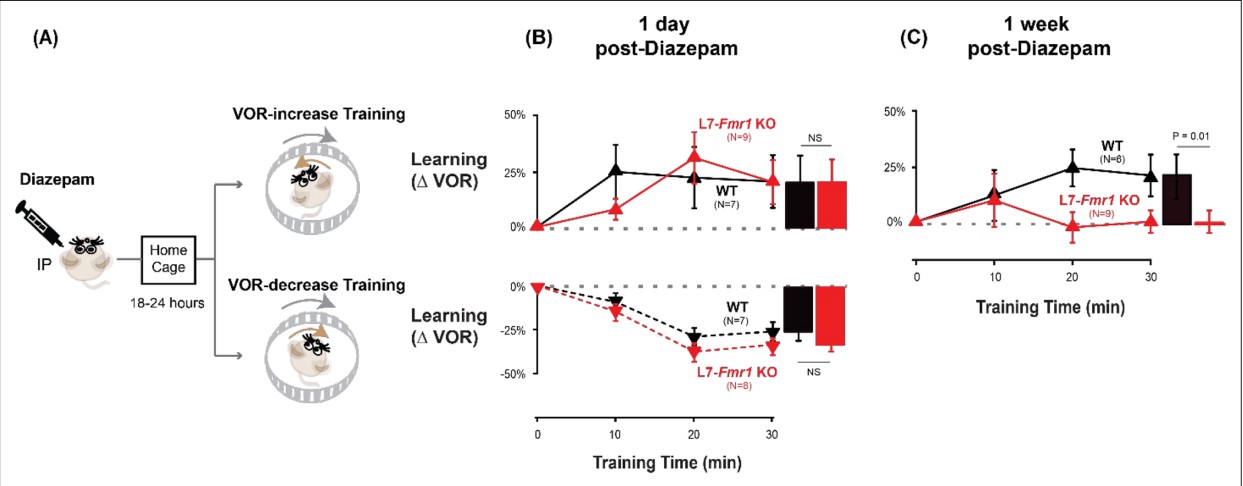

**Figure 3.** Diazepam pre-treatment rescued learning impairment of L7-*Fmr1* KO mice with enhanced associative LTD. (**A**) Mice were given an intraperitoneal (IP) injection of diazepam (0.5 mg/kg) and then returned to the home cage for 18–24 hr, followed by VOR-increase (**top**) or VOR-decrease (*bottom*) training. (**B**) *Top*, VOR-increase learning 1 day (18–24 hr) after diazepam administration in L7-*Fmr1* KO (*red upward triangles*) and WT mice (*black upward triangles*).*Bottom*, VOR-decrease learning (*downward triangles*) 1 day after diazepam. (**C**) VOR-increase learning in the same mice as in (**B**), 1 week after diazepam treatment, and 18–24 hr after IP saline injection.

The online version of this article includes the following source data and figure supplement(s) for figure 3:

**Source data 1.** Diazepam pre-treatment rescued learning impairment of L7-*Fmr1* KO mice with enhanced associative LTD.

**Figure supplement 1.** Diazepam did not affect baseline VOR performance.

**Figure supplement 2.** The acute effect of diazepam was inhibition of VOR-increase learning.

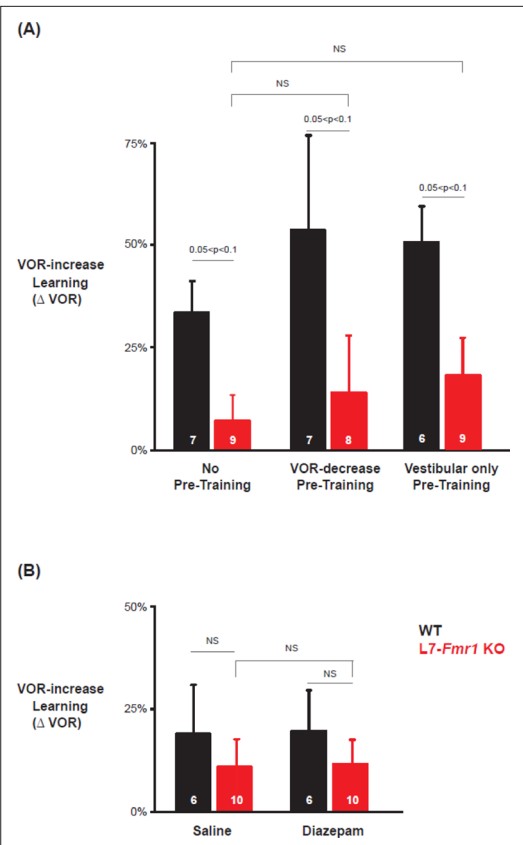

**Figure 4.** Low frequency (0.5 Hz) VOR-increase learning impairment was not rescued by behavioral pre-training or diazepam pre-treatment. (**A**) Low-frequency VOR-increase learning of L7-*Fmr1* KO mice (*red*) and WT mice (*black*), without pre-training (*left*), after 0.5 Hz VOR-decrease pre-training (*middle*), and after 0.5 Hz Vestibular only pre-training (*right*). (**B**) Low frequency (0.5 Hz) VOR-increase learning 18–24 hr after IP injection of saline (*left*) or 0.5 mg/kg diazepam (*right*).

The online version of this article includes the following source data for figure 4:

**Source data 1.** Low frequency (0.5 Hz) VOR-increase learning impairment was not rescued by behavioral pre-training or diazepam pre-treatment.

VOR learning was tested after recovery from this prolonged period of pharmacological suppression of neural activity (*Figure 3A*). Remarkably, the L7-*Fmr1* KO mice exhibited robust VOR-increase learning, comparable to their WT littermates (*Figure 3B*, *top, red* vs. *black*; p=0.86, L7-*Fmr1* KO vs. WT, 30 min, Tukey). Although the same individual L7-*Fmr1* KO had exhibited no significant learning in response to VOR-increase training in the absence of the pharmacological pre-treatment (p=0.99, 0 vs 30 min, data for subset of mice in *Figure 1A* used in *Figure 3* experiments, Tukey), diazepam pre-treatment eliminated this learning deficit.

The enhancement of learning by diazepam pre-treatment was temporary. When the same mice were re-tested one week after diazepam administration, the L7-*Fmr1* KO mice again failed to learn in response to VOR-increase training (*Figure 3C*, *red;* p=0.12, 0 vs 30 min, Tukey). Thus, diazepam pre-treatment could restore the VOR circuit of L7-*Fmr1* KO mice to a state compatible with VOR-increase learning, but this effect was transient.

## Specificity of the effects of pre-training treatments on learning

The ability of behavioral and pharmacological pre-training interventions to enhance learning was specific to mice with enhanced PF-Purkinje cell LTD and to the type of VOR learning task. Wild type mice did not exhibit enhanced VOR-increase learning after diazepam pre-treatment (*Figure 3B top* vs. *Figure 2A*, *black*, p=0.55; *Figure 3B* vs. 3 C, *top, black*, p=0.99; paired sample t-test). Moreover, there was no effect of diazepam pre-treatment on VOR-decrease learning in either the WT or L7-*Fmr1* KO mice (*compare Figure 3B*, *bottom* vs. *Figure 1B*; p=0.91, WT, 1 day post-diazepam vs. control, VOR-decrease at 30 min, paired sample t-test; p=0.37, L7-*Fmr1* KO, 1 day post-diazepam vs. control, VOR-decrease at 30 min, paired sample t-test; p=0.11, L7-*Fmr1* KO vs. WT 1 day post-diazepam, VOR-decrease at 30 min, Tukey). Thus, both the learning impairment in the L7-*Fmr1* KO mice and the effects of diazepam pre-treatment were selective for VOR-increase learning, consistent with previous evidence that this form of VOR learning is more dependent on PF-Purkinje cell LTD than VOR-decrease learning (*Li et al., 1995*; *Boyden et al., 2006*; *Hansel et al., 2006*; *Guo et al., 2014*; *Kimpo et al., 2014*; *Nguyen-Vu et al., 2017*; *Kakegawa et al., 2018*; *Jang et al., 2020*; *Shim et al., 2022*; *Zhang et al., 2023*).

Previous work has also suggested a selective contribution of PF-Purkinje cell LTD to VOR learning induced with high-frequency (≥1 Hz) vestibular and visual stimuli, with less contribution of LTD when VOR learning is induced with low-frequency (≤0.66 Hz) vestibular and visual stimuli (*Boyden et al., 2006*; *Nguyen-Vu et al., 2017*). We found a trend for low-frequency (0.5 Hz, *Figure 4*) as well as high-frequency (1 Hz, *Figures 1–3*) VOR-increase learning to be impaired in the L7-*Fmr1* KO mice (*Figure 4A*, *left, red* vs. *black*; p=0.06, L7-*Fmr1* KO vs. WT, 30 min, Tukey). However, the low-frequency learning impairment was not reversed by the pre-training procedures that reversed

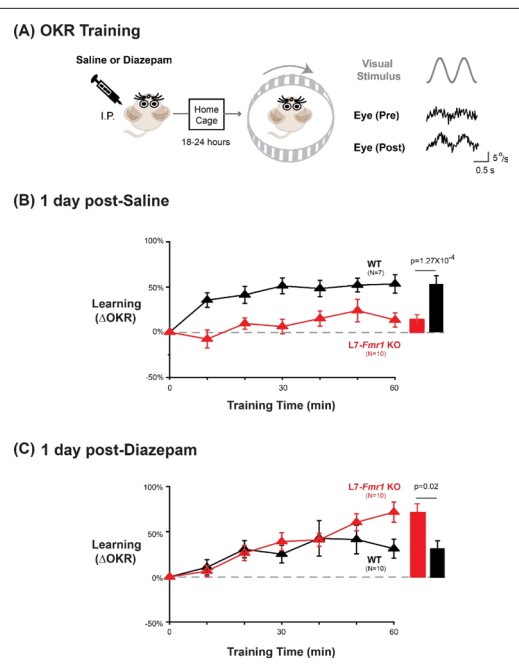

**Figure 5.** Diazepam pre-treatment rescued OKR learning impairment of L7-*Fmr1* KO mice with enhanced associative LTD. (**A**) *Left,* OKR adaptation was assessed 18–24 hr after a single injection of saline or diazepam. *Middle* OKR adaptation was induced by rotating a striped optokinetic drum about an earth-vertical axis centered on the head of the mouse with a 1 Hz sinusoidal velocity profile and peak velocity of ± 10°/s. *Right,* Example raw eye velocity responses (*black*) to the optokinetic visual stimulus (*gray*), measured at the beginning (Pre) and end (Post) of 60 min of OKR adaptation training. (**B**) Average learned change in the amplitude of the OKR relative to pre-training, after each 10 min OKR training block in the L7-*Fmr1* KO (*red*) and WT mice (*black*), 1 day after saline injection. (**C**) Average learned change in the amplitude of the OKR relative to pre-training in L7-*Fmr1* KO (*red*) and WT (*black*) mice that received diazepam (0.5 mg/kg) the previous day.

The online version of this article includes the following source data and figure supplement(s) for figure 5:

**Source data 1.** Diazepam pre-treatment rescued OKR learning impairment of L7-*Fmr1* KO mice with enhanced associative LTD.

**Figure supplement 1.** Baseline optokinetic reflex (OKR) performance normal in L7-*Fmr1* KO mice and after diazepam pre-treatment.

the high-frequency learning impairment. Neither behavioral pre-training (*Figure 4A*, *red, middle,* p=0.47, L7-*Fmr1* KO, VOR-decrease Pre-training vs. no Pre-training; *right,* p=0.35, L7-*Fmr1* KO, Vestibular only Pre-training vs. no Pre-training; paired sample t-test), nor treatment with diazepam 18–24 hr before training (*Figure 4B*, *compare red bars;* p=0.66, L7-*Fmr1* KO, saline vs. diazepam, paired sample t-test), reversed the impairment of low-frequency VOR-increase learning in the L7-*Fmr1* KO mice, in contrast to their effectiveness at reversing the impairment of high-frequency VOR-increase learning (*Figures 2 and 3*). This is consistent with the hypothesis that the behavioral and pharmacological pre-training manipulations selectively restore the capacity for learning tasks that depend on PF-Purkinje cell LTD.

To further test the specificity of the effects of *Fmr1* knockout and diazepam pre-treatment for learning tasks that depend on PF-Purkinje cell LTD, we tested optokinetic reflex (OKR) adaption, which is the oculomotor learning task for which there is arguably the strongest evidence for a critical role of PF-Purkinje cell LTD (*Takeuchi et al., 2008*; *Wang et al., 2014*; *Inoshita and Hirano, 2018*; *Kakegawa et al., 2018*). The OKR was elicited by rotating a striped drum about an earth-vertical axis centered on the head of the mouse, using a sinusoidal velocity profile with frequency of 1 Hz and peak velocity of ± 10°/s (*Figure 5*), with the head restrained and stationary. Baseline OKR gain was not significantly different in the L7-*Fmr1* KO and WT mice (*Figure 5—figure supplement 1*). In WT mice, 60 min of training with the optokinetic stimulus induced a learned increase in the amplitude of the OKR response, as the mice learned to more closely match their eye movements to the motion of the optokinetic stimulus and thus improve stability of its image on the retina (*Figure 5A*, *black,* p=0.001, 0 vs 60 min, Tukey). The L7-*Fmr1* KO mice with enhanced PF-Purkinje cell LTD had significantly impaired OKR adaptation (*Figure 5A*, *red* vs. *black;* p=1.27 × 10⁻⁴, L7-*Fmr1* KO vs. WT, 60 min; p=0.87, L7-*Fmr1* KO, 0 vs 60 min; Tukey). This learning deficit was completely eliminated by pre-treatment with diazepam 18–24 hr before training (*Figure 5B*). Indeed, the L7-*Fmr1* KO mice not only learned better 18–24 hr after diazepam than

they did without this pre-treatment (Compare *Figure 5A* vs. B, *red*; p=0.0001, saline vs. diazepam pre-treatment, 60 min, Tukey), but their learning after diazepam pre-treatment was even enhanced relative to WT (*Figure 5B*, *red* vs. *black*; p=0.02, L7-*Fmr1* KO vs. WT, 60 min, Tukey). Thus, the OKR adaptation results support the findings from high-frequency VOR-increase learning suggesting a

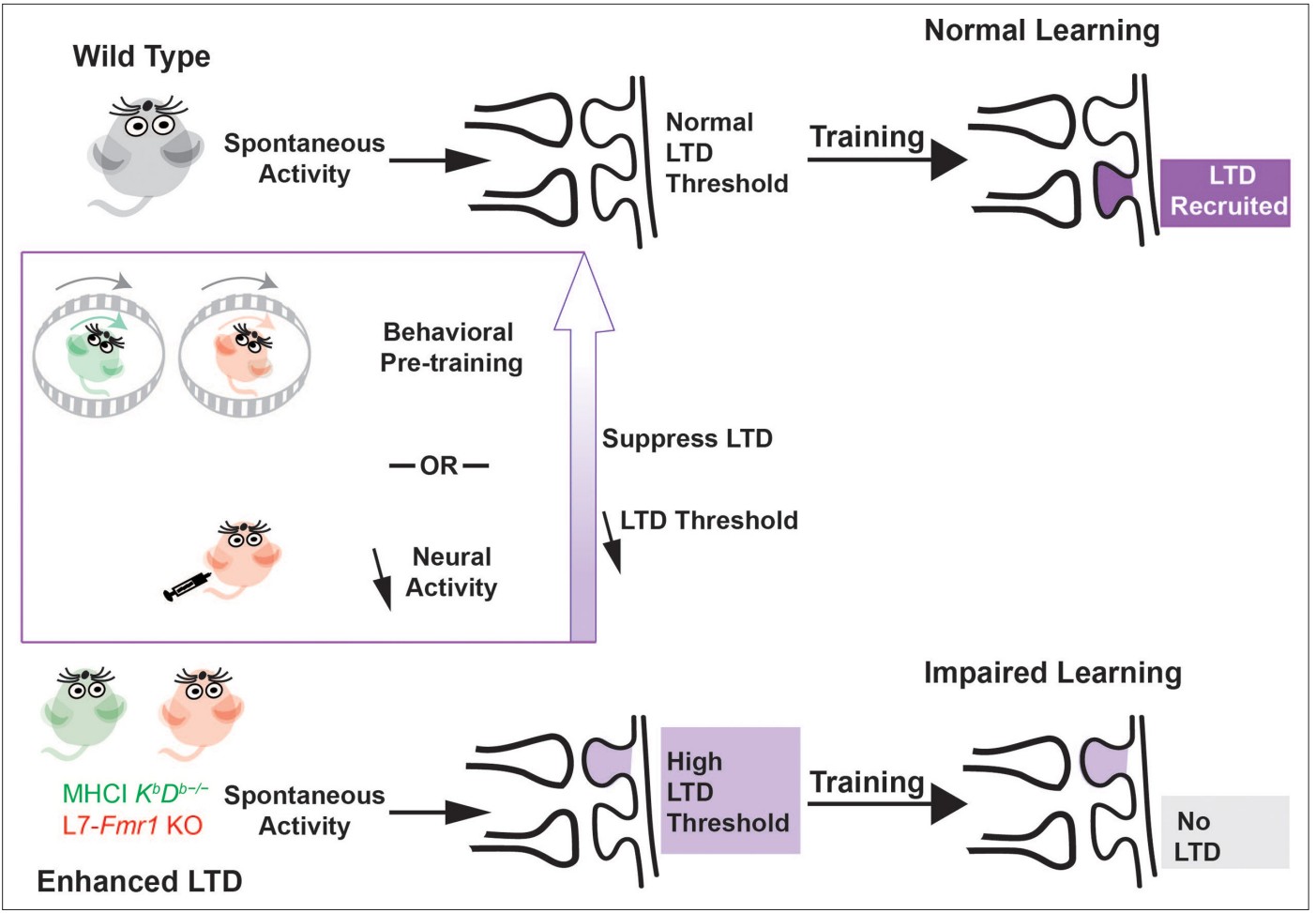

**Figure 6.** Metaplasticity hypothesis for how enhanced plasticity could impair learning. *Top,* In naïve wild type mice, synapses have a normal threshold for associative LTD, and undergo LTD (*dark violet*) in response to training, thereby supporting normal learning. *Bottom,* In mice with enhanced LTD, such as L7-*Fmr1* KO (*pink*) and MHCI K$^b$D$^{b-/-}$ (*green*), the lower threshold for induction of LTD allows it to be aberrantly recruited by spontaneous activity in the circuit (*light violet*), thereby increasing the threshold for additional LTD induction. This prevents the recruitment of LTD during training at the synapses where it is needed to support learning (*grey box*), which impairs learning. Behavioral pre-training or drugs that reduce neural activity can suppress LTD induction and reset the threshold for LTD to normal (*upward purple arrow*), restoring the capacity for LTD-dependent learning.

deficit in PF-Purkinje cell LTD-dependent learning in the mice with enhanced LTD that varies dramatically with the recent history of activity in the circuit.

## Discussion

A question of central scientific and clinical importance is why the enhancement of synaptic plasticity can impair rather than enhance learning. One hypothesis is that a lower threshold for the induction of plasticity might cause it to be over-recruited during training, at synapses that should not have undergone plasticity in addition to synapses where it would support adaptive behavioral changes, thereby corrupting the memory trace (*Migaud et al., 1998*; *Koekkoek et al., 2005*). An alternative hypothesis is that the enhancement of plasticity might allow spontaneous activity in the circuit to aberrantly recruit the plasticity mechanism even before training begins, and thereby reduce its availability during training to support new learning (*Figure 6*). *Nguyen-Vu et al., 2017* described this reduced availability as saturation of plasticity, but more generally, the reduced availability could reflect an increased threshold for the induction plasticity (*Bienenstock et al., 1982*; *Leet et al., 2022*). This threshold metaplasticity hypothesis differs from the over-recruitment hypothesis by suggesting that the enhanced plasticity mechanism is under- rather than over- recruited during training. It also differs by suggesting that the problem with enhanced plasticity arises because of what it does to the circuit

before training, rather than how it functions during training, and therefore more readily accounts for effects of pre-training manipulations on learning in mice with enhanced plasticity. The current findings thus bolster the evidence from *Nguyen-Vu et al., 2017* suggesting threshold metaplasticity rather than over-recruitment as the cause of impaired learning in mice with enhanced associative LTD at the cerebellar parallel fiber-Purkinje cell synapses.

We found that L7-*Fmr1* KO mice with enhanced PF-Purkinje cell LTD exhibit impaired rather than enhanced learning on oculomotor learning tasks in which PF-Purkinje cell LTD has been implicated, as previously shown for MHCI K$^b$D$^{b-/-}$ mice. In both mouse lines, behavioral manipulations designed to prevent or reverse the induction of PF-Purkinje cell LTD during the period before training reversed the learning impairment. The additional finding that pharmacological suppression of neural activity with diazepam can enhance subsequent learning in mice with enhanced associative LTD provides new evidence that these mice are not incapable of LTD-dependent learning. Rather, the interaction of the ongoing neural activity in the circuit with the enhanced plasticity appears to create a state in which LTD is unavailable to support learning. However, this state of reduced availability can be reversed when the patterns of neural activity that create it are eliminated. The capacity for PF-Purkinje cell LTD-dependent learning is dynamically regulated by the recent history of activity.

Comparison across closely related cerebellum-dependent learning tasks reveals the specific behavioral consequences of enhanced PF-Purkinje cell LTD. Although it was the main candidate mechanism of cerebellum-dependent learning for many decades, there is growing evidence that PF-Purkinje cell LTD contributes selectively to certain cerebellum-dependent learning tasks, and not others (*Shibuki et al., 1996*; *Boyden et al., 2006*; *De Zeeuw et al., 2021*). Oculomotor learning is particularly advantageous for analyzing the function of PF-Purkinje cell LTD during learning because this plasticity mechanism is thought to contribute differentially to a set of closely related oculomotor learning tasks that all depend on the same vestibular, visual and motor signaling pathways through the cerebellar flocculus. Despite the shared cerebellar and extra-cerebellar circuitry, a number of experimental approaches, including ex vivo slice physiology (*Inoshita and Hirano, 2018*; *Jang et al., 2020*; *Shim et al., 2022*), optogenetic stimulation (*Kimpo et al., 2014*; *Nguyen-Vu et al., 2017*; *Zhang et al., 2023*) and studies of oculomotor learning in mice with impaired LTD (*Boyden et al., 2006*; *Hansel et al., 2006*; *Kakegawa et al., 2018*) have suggested a selective contribution of PF-Purkinje cell LTD to OKR adaptation and VOR-increase learning induced by high-frequency (≥1 Hz) vestibular and visual stimuli, with less or no contribution to VOR-decrease learning or VOR-increase learning induced with lower frequency vestibular and visual stimuli.

In both the L7-*Fmr1* KO and MHCI K$^b$D$^{b-/-}$ mice with enhanced PF-Purkinje cell LTD, the learning impairments and the enhancement of learning by behavioral or pharmacological pre-training manipulations were remarkably selective for the oculomotor learning tasks in which PF-Purkinje cell LTD has been most strongly implicated. Both lines of mice were profoundly impaired on high-frequency VOR-increase learning and OKR adaptation. The effects of behavioral and pharmacological pre-training were strikingly specific for the same oculomotor learning tasks. Moreover, pre-training only enhanced learning in the L7-*Fmr1* KO and MHCI K$^b$D$^{b-/-}$ mice, and not WT mice, consistent with the pre-training selectively reversing limitations caused by enhanced LTD, rather than generally enhancing cerebellum-dependent learning. In the L7-*Fmr1* KO mice, low-frequency as well as high-frequency VOR-increase learning was impaired. However, the behavioral and diazepam pre-treatments designed to prevent or reverse LTD during the period before training only improved the high-frequency and not the low-frequency VOR-increase learning in the L7-*Fmr1* KO mice, suggesting different mechanistic underpinnings of the low- and high-frequency impairments. In other words, the deletion of *Fmr1* from Purkinje cells may have two distinct effects: enhancement of PF-Purkinje cell LTD, which recapitulates the high-frequency VOR-increase learning phenotypes observed in the MHCI K$^b$D$^{b-/-}$ mice with enhanced LTD, plus disruption of an additional cellular mechanism that contributes to low-frequency VOR-increase learning. Overall, in the two lines of mice with enhanced PF-Purkinje cell LTD, both the learning impairments and the effects of manipulations designed to reverse or prevent LTD before training were remarkably selective for the specific oculomotor learning tasks with the strongest evidence for a contribution of PF-Purkinje cell LTD. This oculomotor learning task specificity strengthens the evidence that the behavioral phenotypes in the L7-*Fmr1* KO and MHCI K$^b$D$^{b-/-}$ mice are a reflection of the enhanced LTD at PF-Purkinje cell synapses rather than other functional properties of the Purkinje cells, their synapses, or other cells or plasticity mechanisms within the same circuit.

The very similar behavioral phenotypes observed when PF-Purkinje cell LTD is enhanced by manipulating different molecular cascades further strengthens the evidence that their shared effect of enhancing LTD is responsible for their shared learning impairments, rather than other, off-target effects of the molecular manipulations. MHCI H2- $D^b$ acts on MAP kinase and integrin via interaction with immune receptors such as PirB (*Shatz, 2009*), whereas *Fmr1* acts by inhibiting mGluR-dependent dendritic protein translation (*Huber et al., 2002*). Cell-type-specific manipulations localize the site where loss of expression of MHCI H2-$D^b$ or *Fmr1* yields the oculomotor learning deficits to the Purkinje cells. Each yields enhancement of PF-Purkinje cell LTD, as measured by the ability of protocols that fail to induce LTD in slices from WT mice to induce LTD in slices from MHCI $K^bD^{b-/-}$ (*McConnell et al., 2009*) and L7-*Fmr1* KO mice (*Koekkoek et al., 2005*). Along with this shared effect of enhancing PF-Purkinje cell LTD, loss of MHCI H2-$D^b$ or *Fmr1* expression from the Purkinje cells may each have additional effects on the intrinsic and synaptic properties of Purkinje cells, which are likely to be different for the two molecules. These latter, 'off-target' effects may contribute to the impairment of L7-*Fmr1* KO mice on low-frequency VOR-increase learning (*Figure 4*), which was not observed in MHCI $K^bD^{b-/-}$ mice (*Figure 1—figure supplement 1*, *Nguyen-Vu et al., 2017*). Off-target effects also might have contributed to the different cerebellar learning phenotypes reported previously in the two lines of mice – enhanced rotorod learning in the MHCI $K^bD^{b-/-}$ mice (*McConnell et al., 2009*), but impaired eyeblink conditioning in the L7-*Fmr1* KO mice, global *Fmr1* KO mice and Fragile X patients (*Koekkoek et al., 2005*). However, the behavioral tasks used in those previous studies were also different, and our current results highlight the importance of the specific choice of behavioral task for assessing cerebellar learning, and the differential dependence of different cerebellar learning tasks on specific molecular and cellular processes within the cerebellum.

The current findings and the metaplasticity conceptual framework could guide the development of new clinical approaches for Fragile X syndrome and a range of other neurological and psychiatric conditions with enhanced associative plasticity. Pre-treatment with the FDA-approved drug diazepam restored the capacity for high-frequency VOR-increase learning and OKR adaptation in the L7-*Fmr1* KO mice without compromising other forms of oculomotor learning or baseline oculomotor performance. This was true even for VOR-decrease learning, which may depend on LTP of the same population of PF-Purkinje cell synapses that undergo LTD during VOR-increase learning (*Shim et al., 2022*). In other words, diazepam pre-treatment fully rescued the learning impairments with no apparent side effects on other, closely related functions of the same neural circuitry. This specificity enhances its therapeutic potential. At the same time, this approach of suppressing neural activity during a pre-training period may be generally applicable to any learning task, motor or cognitive (*Rochefort et al., 2013*; *Badura et al., 2018*; *Ashburn et al., 2020*; *Frontera et al., 2020*; *Stoodley and Tsai, 2021*; *Hwang et al., 2022*), that is impaired by enhanced associative LTD. Pharmacological suppression of neural activity should suppress PF-Purkinje cell LTD throughout the cerebellum, and hence may have the general effect of restoring all regions of the cerebellum to a state compatible with new LTD-dependent learning. This generality of the pharmacological approach stands in contrast to the behavioral pre-training approach, which would require extensive additional knowledge and experimentation to design the appropriate behavioral pre-treatment to reset each functional region of the cerebellum to a state compatible with LTD-dependent learning, and different behavioral pre-treatments would be required to target each of the many functional regions supporting the myriad motor and cognitive functions of the cerebellum. Thus, from a practical standpoint, pharmacological pre-treatment to prevent or reverse the recruitment of LTD before training and thereby lower the threshold for its subsequent recruitment during training is a more feasible and general approach to restoring the capacity for PF-Purkinje cell LTD-dependent learning. Such an approach could be tested even when the specific plasticity mechanism responsible for a learning deficit has not been identified, because lower activity should reduce induction of most associative plasticity mechanisms. Thus, the approach of limiting neural activity during a period before training to reset elevated thresholds for plasticity may be broadly applicable throughout the brain for resetting neural circuits to a state compatible with adaptive plasticity and new learning. Consistent with this, suppression of neural activity in the retina has been successfully employed to reset the visual circuitry and enable recovery from amblyopia in adult mice and cats (*Fong et al., 2021*). The suppression of neural activity may be an especially useful approach if plasticity is pathologically enhanced in areas like the cerebellum or basal ganglia, with a high level of spontaneous spiking activity.

The results predict history-dependent changes in the availability of PF-Purkinje cell LTD to support learning due to activity-dependent changes in the threshold for LTD. Threshold metaplasticity has not been directly documented at these synapses, however several factors that influence climbing fiber-induced calcium influx or the probability of LTD induction have been identified (reviewed in *Zang and De Schutter, 2019*), which could provide the mechanistic substrate for threshold metaplasticity. These include plasticity of the climbing fiber synapse onto the Purkinje cell (*Hansel and Linden, 2000*), plasticity of Purkinje cell dendritic excitability (*Ohtsuki et al., 2012*), changes in the number of spikes in a climbing fiber burst (*Mathy et al., 2009*; *Medina and Lisberger, 2008*), changes in short-term plasticity mechanisms at the PF-Purkinje cell synapses (*Hunley et al., 2023*), and plasticity of inhibitory synapses onto the Purkinje cells (*Kano et al., 1992*; *Kawaguchi and Hirano, 2007*; *Rowan et al., 2018*).

The concept of experience-dependent changes in the threshold for synaptic plasticity has been highly influential in theoretical and computational neuroscience (*Abraham and Bear, 1996*; *Benusková et al., 1999*; *Cooper and Bear, 2012*; *Hulme et al., 2012*; *Lee, 2022*), since the foundational work of *Bienenstock et al., 1982*. However, experimental evidence for whether and how such threshold meta-plasticity supports the function of neural circuits has been limited, and derived largely from studies of the effects of sensory deprivation on the functional connectivity of circuits (*Mioche and Singer, 1989*; *Kirkwood et al., 1996*; *Philpot et al., 2003*; *He et al., 2007*; *Yee et al., 2017*) rather than studies of learning per se. Analysis of threshold metaplasticity in the context of cerebellum-dependent learning and associative LTD offers a new perspective on the Bienenstock, Cooper, and Munro (BCM) model (*Bienenstock et al., 1982*). Most fundamentally, the present results predict threshold metaplasticity at synapses where the plasticity is not Hebbian. A sliding threshold for plasticity has been conceived as a mechanism for countering an instability inherent in Hebbian LTP whereby correlated pre- and post-synaptic activity strengthens a synapse, which leads to an increase in correlated activity, which in turn leads to further strengthening. An increased threshold for LTP in response to an increase in neural activity would counter this instability and provide a mechanism to stabilize firing rates and synaptic weights within a desired range (*van Rossum et al., 2000*; *Toyoizumi et al., 2014*; *Yger and Gilson, 2015*; *Zenke et al., 2017*). In contrast, plasticity at the cerebellar PF-Purkinje cell synapse is described as 'anti-Hebbian' because the associative form of plasticity is LTD. Associative LTD lacks the insta-bility inherent in Hebbian LTP. Moreover, an increased threshold for LTD in response to an increase in neural activity or a decreased threshold for LTD in response to decreased neural activity would tend to oppose rather than support the stability of firing rates of the postsynaptic Purkinje cells. Yet, the present results provide evidence for these activity-dependent changes in the threshold for cerebellar LTD. Thus, rather than supporting homeostatic control of firing rates, the central function of threshold metaplasticity at these synapses may be to limit the amount of plasticity. In addition, the finding that manipulations of neural activity during the pre-training period had different effects in the mice with enhanced LTD than in WT mice suggests that changes in the threshold for plasticity may be driven, not directly by firing rates, but by the recent history of activity-dependent induction of plasticity (*Montgomery and Madison, 2002*; *Lev-Ram et al., 2003*; *Hunley et al., 2023*; *Abraham, 2008*; *Martin and Kosik, 2002*; *Redondo and Morris, 2011*) A plasticity-driven increase in the threshold for further plasticity could serve to protect newly acquired memories from being overwritten (*Fusi et al., 2005*; *Benna and Fusi, 2016*). A second potential function would be to separate memories acquired in close succession onto the synapses of different Purkinje cells, in contrast to findings in the amygdala, where there is evidence that a plasticity-driven decrease in the threshold for further plasticity supports the allocation of memories acquired in close succession to the same neurons (*Han et al., 2007*; *Benna and Fusi, 2016*; *Cai et al., 2016*; *Rashid et al., 2016*; *Lau et al., 2020*).

## Conclusion

We leveraged the relatively simple and well understood physiology and function of the cerebellum and oculomotor system to develop and test a new hypothesis to explain why enhanced plasticity often impairs rather than enhances learning. The current results, along with the previous work by *Nguyen-Vu et al., 2017*, provide convergent evidence that a lower threshold for synaptic plasticity can result in its aberrant recruitment by ongoing activity in a circuit, resulting in an increased threshold for its subse-quent induction and hence the impairment of learning. This threshold metaplasticity perspective may be useful in considering the impact of enhanced plasticity not only in the cerebellum, but in other

brain areas as well, and for developing new clinical approaches for reversing maladaptive plasticity and resetting neural circuits to a state compatible with adaptive plasticity and new learning. More generally, the present results highlight the principle that synaptic properties do not control learning in isolation but interact with the patterns of neural activity in the corresponding circuits to control the capacity for new learning. The implication is that learning deficits associated with abnormal plasticity are not necessarily permanent, but in some cases can be remedied with appropriate reset of the circuit, opening up the possibility for therapeutic approaches targeting neural activity as well as the plasticity mechanisms themselves.

## Materials and methods

All experimental procedures were approved by the Administrative Panel on Laboratory Animal Care (APLAC protocol # 9143) at Stanford University.

### Mice

Mice with the *Fmr1* gene knocked out selectively from cerebellar Purkinje cells were generated through the following breeding strategy. First, homozygous female mice whose *Fmr1* gene, located on the X-chromosome, was floxed (*Fmr1* conditional knockout, cKO; *Mientjes et al., 2006*) were crossed with male mice expressing L7/Pcp2-Cre on an autosome (L7/Pcp2-Cre *Jdhu*; The Jackson Laboratory, Stock No. 010536; *Zhang et al., 2004*). The L7/Pcp2-Cre *Jdhu* line expresses Cre-recombinase in a manner that is highly selective for Purkinje cells. Male offspring from this first cross were mated with females homozygous for the *Fmr1* cKO allele to generate offspring homozygous for *Fmr1* cKO, with some mice L7/Pcp2-Cre-positive and some L7/Pcp2-Cre-negative. Cre-positive offspring of this second cross are referred to as L7-*Fmr1* KO, and their Cre-negative littermates were used as controls and referred to as wild type (WT). Genotyping was performed by Transnetyx Inc on ear-clipped samples to confirm the presence of the floxed *Fmr1* allele in all mice and the presence or absence of Cre using RT-qPCR.

Mice were kept on a reversed 12 hr light/12 hr dark cycle, and behavioral experiments were conducted during the dark cycle of the mice. After surgical implantation (see below), mice were housed individually in standard cages and provided food and water *ad libidum*. Male and female mice 8–22 weeks old were used in the behavioral experiments. Similar learning deficits were observed in male and female L7-*Fmr1* KO mice (*Figure 1—figure supplement 1*), therefore results were pooled across sex.

### Surgery

Mice underwent surgery between 8 and 12 weeks of age to implant hardware for restraining the head and measuring eye movements, as described previously (*Payne and Raymond, 2017*; *Nguyen-Vu et al., 2017*). Mice were anesthetized with 1.5–2.5% isoflurane. An incision was made in the scalp and a custom-made head post (Shapeways Inc) was attached to the top of the skull using dental acrylic (Relyx Unicem Self-Adhesive Universal Resin Cement, Aplicap Capsule Refills-3M). Two stacked neodymium magnets with a total size of 0.75x2 mm (grade N50, axially magnetized, https://super-magnetman.com/) were implanted beneath the conjunctiva on the temporal side of the left eye. An angular magnetic field sensor (HMC1512, Honeywell Inc) was soldered to an 8-pin connector and attached to the skull above the eye using dental acrylic, in a plane parallel to horizontal (nasal-temporal) eye movements. Eye movements were measured by detecting changes in the magnetic field created by movements of the magnet implanted on the eye (*Payne and Raymond, 2017*). Mice recovered from surgery for at least five days before experiments were performed.

### Behavioral experiments

Mice were acclimatized to the laboratory for at least 20 min after transport from the animal care facility before the start of an experiment. Experiments were conducted in a light-proof, sound-attenuated chamber (IAC acoustics). The head of the mouse was secured by attaching its head post to a restrainer, which was then attached to a vestibular turntable controlled by a Carco Model 823 rate table and Model 405D controller. The turntable delivered vestibular stimuli to the mouse by rotation about a yaw (earth-vertical) axis centered on the head of the mouse. An optokinetic drum controlled by a

Yaskawa AC-Servo SGMCS-02B3B11 motor provided visual stimulation by rotation about an earth-vertical axis aligned with that of the vestibular turntable. The drum was made of translucent white plastic, and had alternating black and white stripes, with each stripe subtending approximately 7.5° of the visual field, illuminated by an LED light strip attached to the rim of the drum. Eye movements were recorded using the method described in *Payne and Raymond, 2017*.

Experiments to assess VOR learning consisted of testing blocks and training blocks. Testing blocks consisted of three 45 s tests of the eye movement response to the vestibular stimulus delivered alone in complete darkness, that is, the VOR. The vestibular stimulus was sinusoidal vestibular turntable rotation at 1 Hz or 0.5 Hz with a peak velocity of ± 10°/s. The three 45 s VOR tests in a block were separated by 10 s with the turntable stationary. Training blocks were ten minutes long, and were repeated three times for a total of 30 min training, with a testing block following each training block. For VOR-increase training, the vestibular stimulus used for testing the VOR (1 Hz or 0.5 Hz) was paired with oppositely directed motion of the illuminated optokinetic drum with the same peak velocity (±10°/s). For VOR-decrease training, the vestibular stimulus used for testing was paired with motion of the optokinetic drum in the same direction with the same velocity, so that the drum was stationary relative to the head of the mouse. In behavioral pre-training experiments, the pre-training consisted of three 10 min blocks of either VOR-decrease training or delivery of the vestibular stimulus alone in the dark (Vestibular only), with a testing block before each training block. Calibration of the signals from the magnetic sensor used to record eye movements was performed after the experiment, as described in *Payne and Raymond, 2017*.

Experiments to assess OKR adaptation consisted of sixty 50 s long blocks of training with 1 Hz sinusoidal rotation of the optokinetic drum with peak velocity of ± 10°/s, with the vestibular turntable stationary. Each 50 s block of training was followed by 10 s in darkness.

Prior to some experiments (*Figure 3*, *Figure 3—figure supplements 1 and 2*, *Figure 4*, *Figure 5*, *Figure 5—figure supplement 1*), mice received a single IP injection of 0.4, 0.5, or 2.5 mg/kg diazepam (in saline) or saline control. After diazepam or saline administration, mice were returned to the home cage, and then behavioral testing was performed either 2 hr, 18–24 hr, and/or 1 week later.

Each mouse underwent multiple behavioral experiments, with at least two days between successive experiments. The same cohort of mice was used for the experiments shown in *Figures 1 and 2*, with the order of the experiments randomized. A subset of the same cohort was then used for the diazepam experiments shown in *Figure 3*. A separate cohort of mice was used for the low-frequency training experiments shown in *Figure 4*, with the order of randomized for the behavioral pre-training conditions shown in *Figure 4A* (no pretraining, VOR-decrease pre-training and Vestibular only pre-training) followed by the diazepam pre-treatment experiments in *Figure 4B*, with randomized order for drug and saline conditions. Another separate cohort of mice was used for the OKR adaptation experiments shown in *Figure 5*. The order of experiments with diazepam and saline treatment was pseudorandomized. The experimenters collecting behavioral data were blinded to the genotype and the diazepam versus saline treatment.

## Analysis of eye movement measurements

Signals from the magnetic sensor related to eye position were fourth-order low-pass (15 Hz) Butterworth filtered and then digitally differentiated to obtain eye velocity using a windowed (30 ms) Savitzky-Golay filter. Eye velocity data from each VOR test or OKR block were fit with a 1 Hz or 0.5 Hz sinusoid. Values deviating from the sinusoidal fit by more than 31°/s were identified as saccades or movement artifacts and excluded, along with data from 50 ms before and after. Segments of data less than 10ms in duration were also excluded. The entire 45 s VOR or 50 s OKR test was excluded if more than 45% of the data points were excluded, Subsequently, the remaining eye velocity data underwent a second round of fitting using sinusoids at frequencies of either 1 Hz or 0.5 Hz. The amplitude of this second sinusoidal fit provided the measure of the amplitude of the eye movement response, Values from the three VOR tests in a block were averaged. VOR learning (ΔVOR) was calculated as the percentage change in the VOR amplitude following each 10 min block of training relative to the baseline VOR amplitude measured before training. For OKR, values from the first three 50 s OKR blocks were averaged to obtain the pre-training baseline, and values from the last three blocks (58–60 min) were averaged to obtain the post-training OKR measure. OKR learning (ΔOKR) was calculated as the percentage change in the OKR amplitude post- vs. pre-training. Learned changes

in the OKR relative to pre-training are also reported for blocks 10, 20, 30, 40, and 50. Eye movement gain was calculated as the ratio of eye movement amplitude to either vestibular (VOR) or visual (OKR) stimulus amplitude.

## Statistical analysis

Data were analyzed with a Shapiro-Wilk test of normality, followed by a two-factor repeated measures ANOVA with posthoc Tukey or by a two-sample or paired sample t-test, executed in OriginPro 2022 software. A value of p less than 0.05 was considered significant. Data are plotted as mean ± SEM.

## Acknowledgements

We thank David Nelson's group at Baylor College of Medicine for providing *Fmr1* cKO mice. We thank Sriram Jayabal for his technical advice, Macarena Martinez Rey and Daniel Geimer for their support in managing the mouse colony, Maxwell Gagnon and Brian Angeles for writing code for data acquisition and analysis, and all members of the JLR laboratory for their support and input. In preparing this manuscript, we made a conscious effort to address citation bias. Following the approach outlined in Dworkin et al., 2020, we used open source code to assess the gender balance of our citations based on the first names of the first and last authors (Zhou et al., 2020). Excluding self-citations, our article includes citations as follows: 56.98% man/man, 9.30% man/woman, 23.26% woman/man, and 10.47% woman/woman citations. For comparison, the proportions obtained from articles in the top five neuroscience journals (Dworkin et al., 2020) are as follows: 58.4% man/man, 9.4% man/woman, 25.5% woman/man, and 6.7% woman/woman. Our references also contain 21.96% author of color (first)/author of color(last), 18.55% white author/author of color, 21.03% author of color/white author, and 38.46% white author/white author.

## Additional information

### Competing interests

Jennifer L Raymond: Reviewing editor, eLife. The other authors declare that no competing interests exist.

### Funding

| Funder | Grant reference number | Author |
|---|---|---|
| National Institutes of Health | R01 DC004154 | Jennifer L Raymond |
| National Institutes of Health | R01 NS072406 | Jennifer L Raymond |
| Simons Foundation | 543031 | Jennifer L Raymond |
| Stanford University | Dean's Fellowship | Amin MD Shakhawat |
| National Eye Institute | R01 EY031972 | Jennifer L Raymond |

The funders had no role in study design, data collection and interpretation, or the decision to submit the work for publication.

### Author contributions

Amin MD Shakhawat, Conceptualization, Data curation, Formal analysis, Funding acquisition, Investigation, Visualization, Writing – original draft, Writing – review and editing; Jacqueline G Foltz, Adam B Nance, Jaydev Bhateja, Investigation; Jennifer L Raymond, Conceptualization, Funding acquisition, Writing – original draft, Writing – review and editing

### Author ORCIDs

Amin MD Shakhawat ⓘ https://orcid.org/0000-0002-3776-3699
Jennifer L Raymond ⓘ https://orcid.org/0000-0002-8145-747X

### Ethics

All experimental procedures were approved by the Administrative Panel on Laboratory Animal Care (APLAC protocol # 9143) at Stanford University.

Reviewer #1 (Public review): https://doi.org/10.7554/eLife.92543.3.sa1
Reviewer #2 (Public review): https://doi.org/10.7554/eLife.92543.3.sa2
Author response https://doi.org/10.7554/eLife.92543.3.sa3

---

## Additional files

### Supplementary files

• MDAR checklist

### Data availability

All code used for data acquisition (https://github.com/RaymondLab/Code/tree/Master/Experiment%20Protocols) and analysis (https://github.com/RaymondLab/Code/tree/Master/Tools/VOR_Analysis) is available at https://github.com/RaymondLab/Code (copy archived at *RaymondLab, 2024*). Underlying data for each figure is available in the corresponding Excel file.

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
