## [Editor Report · eLife assessment]

This **important** manuscript follows up on previous findings from the same lab supporting the idea that deficits in learning due to enhanced synaptic plasticity are due to saturation effects. **Compelling** evidence is presented that behavioral learning deficits associated with enhanced synaptic plasticity in a transgenic mouse model can be rescued by manipulations designed to reverse the saturation of synaptic plasticity. In particular, the finding that a previously FDA-approved therapeutic can rescue learning could provide new insights for biologists, psychologists, and others studying learning and neurodevelopment.

---

## [Referee Report · Reviewer #1 (Public review)]

Summary:

Shakhawat et al., investigated how enhancement of plasticity and impairment could result in the same behavioral phenotype. The authors tested the hypothesis that learning impairments result from saturation of plasticity mechanisms and had previously tested this hypothesis using mice lacking two class I major histocompatibility molecules. The current study extends this work by testing the saturation hypothesis in a Purkinje-cell (L7) specific Fmr1 knockout mouse mice, which have enhanced parallel fiber-Purkinje cell LTD. The authors found that L7-Fmr1 knockout mice are impaired on an oculomotor learning task and both pre-training, to reverse LTD, and diazepam, to suppress neural activity, eliminated the deficit when compared to controls.

Strengths:

This study tests the "saturation hypothesis" to understand plasticity in learning using a well-known behavior task, VOR, and an additional genetic mouse line with a cerebellar cell-specific target, L7-Fmr1 KO. This hypothesis is of interest to the community as it evokes novel inquisition into LTD that has not been examined previously.

Utilizing a cell-specific mouse line that has been previously used as a genetic model to study Fragile X syndrome is a unique way to study the role of Purkinje cells and the Fmr1 gene. This increases the understanding in the field in regards to Fragile X syndrome and LTD.

The VOR task is a classic behavior task that is well understood, therefore using this metric is very reliable for testing new animal models and treatment strategies. The effects of pretraining are clearly robust and this analysis technique could be applied across different behavior data sets.

The rescue shown using diazepam is very interesting as this is a therapeutic that could be used in clinical populations as it is already approved.

All previous comments have been addressed with additional studies, explanations, or analyses. These additions strengthen a very impactful study.

The authors achieved their study objectives and the results strongly support their conclusion and proposed hypothesis. This work will be impactful on the field as it uses a new Purkinje-cell specific mouse model to study a classic cerebellar task. The use of diazepam could be further analyzed in other genetic models of neurodevelopmental disorders to understand if effects on LTD can rescue other pathways and behavior outcomes.

---

## [Referee Report · Reviewer #2 (Public review)]

This manuscript explores the seemingly paradoxical observation that enhanced synaptic plasticity impairs (rather than enhances) certain forms of learning and memory. The central hypothesis is that such impairments arise due to saturation of synaptic plasticity, such that the synaptic plasticity required for learning can no longer be induced. A prior study provided evidence for this hypothesis using transgenic mice that lack major histocompatibility class 1 molecules and show enhanced long-term depression (LTD) at synapses between granule cells and Purkinje cells of the cerebellum. The study found that a form of LTD-dependent motor learning-increasing the gain of the vestibulo-ocular reflex (VOR)-is impaired in these mice and can be rescued by manipulations designed to "unsaturate" LTD. The present study extends this line of investigation to another transgenic mouse line with enhanced LTD, namely, mice with the Fragile X gene knocked out. The main findings are that VOR gain increase learning is selectively impaired in these mice but can be rescued by specific manipulations of visuomotor experience known to reverse cerebellar LTD. Additionally, the authors show that a transient global enhancement of neuronal inhibition also selectively rescues gain increase learning. This latter finding has potential clinical relevance since the drug used to boost inhibition, diazepam, is FDA-approved and commonly used in the clinic. The evidence provided for the saturation is somewhat indirect because directly measuring synaptic strength in vivo is technically difficult. Nevertheless, the experimental results are solid. In particular, the specificity of the effects to forms of plasticity previously shown to require LTD is remarkable.

---

## [Author Response]

The following is the authors’ response to the original reviews.

**eLife assessment**
This important manuscript follows up on previous findings from the same lab supporting the idea that deficits in learning due to enhanced synaptic plasticity are due to saturation effects. Compelling evidence is presented that behavioral learning deficits associated with enhanced synaptic plasticity in a transgenic mouse model can be rescued by manipulations designed to reverse the saturation of synaptic plasticity. In particular, the finding that a previously FDA-approved therapeutic can rescue learning could provide new insights for biologists, psychologists, and others studying learning and neurodevelopment.

eLife assessment, Significance of findings

This valuable manuscript follows up on previous findings from the same lab supporting the idea that deficits in learning due to enhanced synaptic plasticity are due to saturation effects.

According to the eLife criteria for assessing significance, the “valuable” assessment indicates “findings that have theoretical or practical implications for a subfield.” We have revised the manuscript to emphasize the “theoretical and practical implications beyond a single subfield” which “substantially advance our understanding of major research questions”, with “profound implications” and the potential for “widespread influence,” the eLife criteria for a designation of “landmark” significance.

The most immediate implications of our results are for the two major neuroscience subfields of cerebellar research and autism research. However, as recognized by Reviewer 2, the implications are much broader than that: “the finding that a previously FDA-approved therapeutic can rescue learning could provide important new insights for biologists, psychologists, and others studying learning and neurodevelopment.” We have substantially revised the Discussion section of the manuscript to more explicitly lay out how the central idea of our manuscript-- that the capacity for learning at any given moment is powerfully influenced by dynamic, activity- and plasticity-dependent changes in the threshold for synaptic plasticity over short timescales of tens of minutes to hours --has implications for scientific thinking and experiments on plasticity and learning throughout the brain, as well as clinical practice for a wide array of brain disorders associated with altered plasticity and learning impairment.

To emphasize the broad conceptual implications of our research, we have reframed our conclusions in terms of metaplasticity rather than saturation of plasticity throughout the revised manuscript. In our previous submission, we had used the “saturation “ terminology for continuity with our previous NguyenVu et al 2017 eLife paper, and mentioned the related idea of threshold metaplasticity in a single sentence: “Similarly, the aberrant recruitment of LTD before training may lead, not to its saturation per se, but to some other kind of reduced availability, such as an increased threshold for its induction (Bienenstock, Cooper, and Munro, 1982; Leet, Bear, and Gaier, 2022).” However, we now appreciate that metaplasticity is a more general conceptual framework for our findings, and therefore emphasize this concept in the revised manuscript, while still making the conceptual link with the “saturation” idea presented in NguyenVu et al 2017 (lines 236-238).

The concept of a sliding threshold for synaptic plasticity (threshold metaplasticity) was proposed four decades ago by Bienenstock, Cooper and Munro (1982) as a mechanism for countering an instability inherent in Hebbian plasticity whereby correlated pre- and post-synaptic activity strengthens a synapse, which leads to an increase in correlated activity, which in turn leads to further strengthening. To counter this, BCM proposed a sliding threshold whereby increases in neural activity increase the threshold for LTP and decreases in activity decrease the threshold for LTP, thereby providing a mechanism for stabilizing firing rates and synaptic weights. This BCM sliding threshold model has been highly influential in theoretical and computational neuroscience, but experimental evidence for whether and how such a mechanism functions in vivo has been quite limited.

Our work extends the previous, limited experimental evidence for a BCM-like sliding threshold in vivo in several significant ways, which we now discuss in the revised manuscript:

First, we analyze threshold metaplasticity at synapses where the plasticity is not Hebbian and lacks the inherent instability that inspired the BCM model. The synapses onto cerebellar Purkinje cells have been described as “anti-Hebbian” because the associative form of plasticity is synaptic LTD of excitatory inputs. This anti-Hebbian associative plasticity lacks the instability inherent in Hebbian plasticity. Moreover, a BCM-like sliding threshold that increases the threshold for associative LTD with increased firing rates and decreases threshold for LTD with decreased firing rates would tend to oppose rather than support the stability of firing rates, nevertheless we find evidence for this in our experimental results. Thus, for cerebellar LTD, the central function of the sliding threshold may not be the stabilization of firing rates, but rather to limit plasticity in order to suppress the overwrite of new memories or to allocate different memories to the synapses of different Purkinje cells.

Second, we analyze the influence of a BCM-like sliding threshold for plasticity on behavioral learning. Most previous evidence for the BCM model in vivo has derived from studies of the effects of sensory deprivation (e.g., monocular occlusion) on the functional connectivity of sensory circuits (Kirkwood et al., 1996; Desai et al. 2002; Fong et al., 2021) rather than on learning per se.

Third, our results provide evidence for major changes in the threshold for plasticity over short time scales and with more subtle manipulations of neural activity than used in previous studies, with practical implications for clinical application. Previously, metaplasticity has been demonstrated with sensory deprivation over multiple days (Kirkwood et al., 1996; Desai et al. 2002) or with drastic changes in neural activity, such as with TTX in the retina (Fong et al, 2021), TMS (Hamada et al 2008), or high frequency electrical stimulation in vitro (Holland & Wagner 1998; Montgomery & Madison 2002) or in vivo (Abraham et al 2001). In contrast, we provide evidence for metaplasticity induced by 30 min of behavioral manipulation (pre-training) and by the relatively subtle pharmacological manipulation of activity with systemic administration of diazepam, a drug approved for humans. Thus, our work contributes not only conceptually to understanding the function of threshold metaplasticity in vivo, but also offers practical observations that could pave the way for novel therapeutic interventions.

Fourth, whereas efforts to enhance plasticity and learning have largely focused on increasing the excitability of neurons during learning to help cross the threshold for plasticity (e.g., Albergaria et al., 2018; Yamaguchi et al., 2020; Le Friec et al., 2017), we take the opposite, somewhat counterintuitive approach of inhibiting the excitability of neurons during a period before learning to reset the threshold for plasticity to a state compatible with new learning. To our knowledge, the only other application of such an approach in an animal model of a brain disorder has been inhibiting peripheral (retinal) activity with TTX for treatment of amblyopia (Fong et al, 2021). Our findings from CNS inhibition with a single systemic dose of diazepam greatly expands the potential applications, which could readily be tested in other mouse models of human disorders, and other learning deficits. Even in cases where the specific synaptic impairments and circuitry are less fully understood, the impact of suppressing neural activity during a period before training to reduce the threshold for plasticity could be empirically tested.

Fifth, our work extends the consideration of a BCM-like sliding threshold for plasticity to the cerebellum, whereas previous work has focused on models and experimental studies of forebrain circuits. Currently there is a surge of interest in the contribution of the cerebellum to functions and brain disorders previously ascribed to forebrain, hence we anticipate broad interest in this work.

Sixth, our results suggest that the history of plasticity rather than the history of firing rates may be the homeostat controlling the threshold for plasticity, at least at the synapses under consideration. Diazepam pre-treatment only enhanced learning in the L7-Fmr1 KO mice with a low “baseline” threshold for plasticity, as measured in vitro, and not WT mice. This suggests it is not the neural activity per se that drives the change in threshold for plasticity, but the interaction of activity with the plasticity mechanism.

In the revised Discussion, we make all of the above points, to make the implications more clear to readers.

The broad interest in this topic is illustrated by two concrete examples. First, an abstract of this work was honored with selection for oral presentation at the November 2023 Symposium of the Molecular and Cellular Cognition Society, a conceptually wide-ranging organization with thousands of members worldwide. Second, the most closely related published work on activity-dependent metaplasticity in vivo, the Fong et al 2021 eLife paper demonstrating reversal of amblyopia by suppression of activity in the retina by TTX, attracted such broad interest, not just of professional scientists, but also the general public, as to be reported on National Public Radio’s All Things Considered, with an audience of 11.9 million people worldwide.

In considering the potential of this work for widespread influence, it is important to note that activitydriven changes in the threshold for plasticity could very well be a general property of most if not all synapses, yet very little is known about its function in vivo, especially during learning. Therefore, the seminal conceptual and practical advances described above have the potential for profound implications throughout neuroscience, psychiatry, neurology and computer science/AI, the eLife criterion for designation as “landmark” in significance. We respectfully request that the reviewers and editor reassess the significance of our findings in light of our much-improved discussion of the broad significance of the work.

eLife assessment, Strength of support

Convincing evidence is presented that behavioral learning deficits associated with enhanced synaptic plasticity in a transgenic mouse model can be rescued by manipulations designed to reverse the saturation of synaptic plasticity. In particular, the finding that a previously FDA-approved therapeutic can rescue learning could provide important new insights for biologists, psychologists, and others studying learning and neurodevelopment.

The designation of “Convincing” indicates “methodology in line with current state-of the-art.” In the revised Discussion, we more clearly highlight that our evidence is “more rigorous than current state-ofthe-art” in several respects, thereby meeting the eLife criterion for “Compelling”:

(1) Comparison of learning deficits and effects of behavioral and pharmacological pretreatment across five closely related oculomotor learning tasks, which all depend on the same region of the cerebellum (the flocculus), but which previous work has found to vary in their dependence on LTD at the cerebellar parallel fiber-to-Purkinje cell synapses.

The “state-of-the-art” behavioral standard in the field of learning is assessment of a single learning task that depends on a given brain area, with the implicit or explicit assumption that the task chosen is representative of “cerebellum-dependent learning” or hippocampus-, amygdala-, basal ganglia-, cortex- dependent learning, etc. Sometimes there is a no-learning behavioral control.

Our study exceeds this standard by comparing across many different closely related learning tasks, which all depend on the cerebellar flocculus and other shared vestibular, visual, and oculomotor circuitry, but vary in their dependence on LTD at the cerebellar parallel fiber-to-Purkinje cell synapses. In the original submission, we reported results for high-frequency VOR-increase learning that were dramatically different than for three other VOR learning tasks for which there is less evidence for a role of LTD. Reviewer 2 noted, “the specificity of the effects to forms of plasticity previously shown to require LTD is remarkable.” In the revised manuscript, we provide new data for a second oculomotor learning task in which LTD has been implicated, OKR adaptation, with very similar results as for high-frequency VORincrease learning. The remarkable specificity of both the learning deficits and the effects of pre-training manipulations, in two different lines of mice, for the two specific learning tasks in which LTD has been most strongly implicated, and not the other three oculomotor learning tasks, substantially strengthens the evidence for the conclusion that the learning deficits and effects of pre-training are related specifically to the lower threshold for LTD, rather than the result of some other effect of the gene KO or pre-treatment on the cerebellar or oculomotor circuitry (discussed on lines 270-290 of revised manuscript).

(2) Replication of findings in more than one line of mice, targeting distinct signaling pathways, with a common impact of enhancing LTD at the cerebellar PF-Purkinje cell synapses.

State-of-the-art is to report the effects of one specific molecular signaling pathway on behavior.

In the first part of this Research Advance, we replicate the findings of Nguyen-Vu et al 2017 for a completely different line of mice with enhanced LTD at the parallel fiber-to-Purkinje cell synapses. Like the comparison across LTD-dependent and LTD-independent oculomotor learning tasks, the comparison across completely different lines of mice with enhanced LTD strengthens the evidence that the shared behavioral phenotypes are a reflection of the state of LTD rather than other “off-target” effects of each mutation (discussed on lines 291-309 of revised manuscript).

(3) Reversal of learning impairments with more than one type of treatment.

State-of-the-art is to be able to reverse a learning deficit or other functional impairment in an animal model of a brain disorder with a single treatment; indeed, success in this respect is viewed as wildly exciting, as evidenced by the reception by the scientific and lay communities of the Fong et al, 2021 eLife report of reversal of amblyopia by TTX treatment of the retina.

In the current work, we demonstrate reversal of learning deficits with two different types of treatment during the period before training, one behavioral and one pharmacological. The current diazepam pretreatment results provide a fundamentally new type of evidence for the hypothesis that the threshold for LTD and LTD-dependent learning varies with the recent history of activity in the circuit, complementing the evidence from behavioral and optogenetic pre-training approaches used previously in Nguyen-Vu et al, 2017 (discussed on lines 151-158 and 246-255 of revised manuscript).

**Public Reviews:**

**Reviewer #1 (Public Review):**
Summary:Shakhawat et al., investigated how enhancement of plasticity and impairment could result in the same behavioral phenotype. The authors tested the hypothesis that learning impairments result from saturation of plasticity mechanisms and had previously tested this hypothesis using mice lacking two class I major histocompatibility molecules. The current study extends this work by testing the saturation hypothesis in a Purkinje-cell (L7) specific Fmr1 knockout mouse mice, which have enhanced parallel fiber-Purkinje cell LTD. The authors found that L7-Fmr1 knockout mice are impaired on an oculomotor learning task and both pre-training, to reverse LTD, and diazepam, to suppress neural activity, eliminated the deficit when compared to controls.Strengths:This study tests the "saturation hypothesis" to understand plasticity in learning using a well-known behavior task, VOR, and an additional genetic mouse line with a cerebellar cell-specific target, L7-Fmr1 KO. This hypothesis is of interest to the community as it evokes a novel inquisition into LTD that has not been examined previously.Utilizing a cell-specific mouse line that has been previously used as a genetic model to study Fragile X syndrome is a unique way to study the role of Purkinje cells and the Fmr1 gene. This increases the understanding in the field in regards to Fragile X syndrome and LTD.The VOR task is a classic behavior task that is well understood, therefore using this metric is very reliable for testing new animal models and treatment strategies. The effects of pretraining are clearly robust and this analysis technique could be applied across different behavior data sets.The rescue shown using diazepam is very interesting as this is a therapeutic that could be used in clinical populations as it is already approved.There was a proper use of controls and all animal information was described. The statistical analysis and figures are clear and well describe the results.

We thank the reviewer for summarizing the main strengths of our original submission. We have further strengthened the revised submission by

(1) more fully discussing the broad conceptual implications, as outlined above;

(2) adding additional new data (Fig. 5) showing that another LTD-dependent oculomotor learning task, optokinetic reflex (OKR) adaptation, is impaired in the L7-Fmr1 KO mice and rescued by pre-treatment with diazepam, as we had already shown for high-frequency VOR increase learning; (3) responding to the specific points raised by the reviewers, as detailed below.

Weaknesses:While the proposed hypothesis is tested using genetic animal models and the VOR task, LTD itself is not measured. This study would have benefited from a direct analysis of LTD in the cerebellar cortex in the proposed circuits.

Our current experiments were motivated by the direct analysis of cerebellar LTD in Fmr1 knock out mice that was already published (Koekkoek et al., 2005). In that previous work, LTD was analyzed in both Purkinje cell selective L7-Fmr1 KO mice (Koekkoek et al., 2005; Fig. 4D), as used in our study, and global Fmr1 knock out mice (Koekkoek et al., 2005; Fig. 4B). Both lines were found to have enhanced LTD, as cited in the Introduction of our manuscript (lines 48-51, 63-64). The goal of our current study was to build on this previous work by analyzing the behavioral correlates of the findings from this previous, direct analysis of LTD.

Diazepam was shown to rescue learning in L7-Fmr1 KO mice, but this drug is a benzodiazepine and can cause a physical dependence. While the concentrations used in this study were quite low and animals were dosed acutely, potential side-effects of the drug were not examined, including any possible withdrawal.

In humans, diazepam (valium) is one of the most frequently prescribed drugs in the world, and the side effects and withdrawal symptoms have been extensively studied and documented.1 Withdrawal symptoms are generally not observed with treatments of less than 2 weeks (Brett and Murnion, 2015). After longterm treatments tapering of the dosage is recommended to mitigate withdrawal (Brett and Murnion, 2015 and https://americanaddictioncenters.org/valium-treatment/withdrawal-duration). The extensive data on the safety of diazepam in humans lowers the barrier to potential clinical translation of our basic science findings, although we emphasize that our own expertise is scientific, and translation to Fragile X patients or other patient groups will require additional development of the research by clinicians.

Given the extensive history of research on this drug, we focused on looking for side effects that would reflect an adverse effect of diazepam on the function of the same oculomotor neural circuitry whose ability to support certain oculomotor learning tasks was improved after diazepam. In other words, we assessed whether the pharmacological manipulation was enhancing certain functions of a given circuit at the expense of others. As we note (line 164), “The acute effect of diazepam administration [measured 2 hours after administration] was to impair learning” in both WT and L7-Fmr1 KO mice. One could consider this a side effect. More importantly, we also tested extensively for oculomotor side-effects during the therapeutic period when learning impairments were eliminated in the L7-Fmr1 KOs, 18-24 hours post-administration, and have a full section of the Results describing our findings about this, titled “Specificity of pre-training effects on learning.” As described in the Results and Discussion (lines 184195, 312-318, Figure 3, figure 3-supplement1; figure 4B; figure 5-supplement 1), we found no such adverse side-effects, which is again encouraging with respect to the translational potential of our findings.

This drug is not specific to Purkinje cells or cerebellar circuits, so the action of the drug on cerebellar circuitry is not well understood for the study presented.

The effects of diazepam are indeed not specific to Purkinje cells, but rather are known to be widespread. Diazepam is a positive allosteric modulator of GABAA receptors, which are found throughout the brain, including the cerebellum. When delivered systemically, as we did in our experiments, diazepam will suppress neural activity throughout the brain by facilitating inhibition, as documented by decades of previous research with this and related benzodiazepines, including dozens of studies of the effects of diazepam in the cerebellum.

To our knowledge, there is currently no drug that can specifically inhibit Purkinje cells, especially one that can be given systemically to cross the blood-brain barrier. Moreover, if such a drug did exist, we would not predict it to have the same effect as diazepam in reversing the learning deficits of the L7-Fmr1 KO mice, because the latter presumably depends on suppression of activity in the cerebellar granule cells and neurons of the inferior olive, whose axons form the parallel fibers and climbing fibers, and whose correlated activity controls LTD at the parallel fiber-Purkinje cell synapses.

We have revised the text to clarify the key point that despite its widespread action on the brain, the effects of diazepam on cerebellum-dependent learning were remarkably specific (lines 184-195, 210-228, 312318). During the period 18-24 hours after a single dose of diazepam, the learning deficits of L7-Fmr1 KO mice on two LTD-dependent oculomotor learning tasks were completely reversed, with no effects on the same tasks in WT mice, and no effects (“side-effects”) in L7-Fmr1 KO mice or WT mice on other, LTDindependent oculomotor learning tasks that depend on the same region of the cerebellum, and no effects on baseline performance of visually or vestibularly driven eye movements.

As described in the revised Discussion (lines 318-323), the non-specific mild suppression of neural activity throughout the brain by diazepam makes it a potentially generalizable approach for inducing BCM-like shifts in the threshold for associative plasticity to facilitate subsequent learning. More specifically, diazepam-mediated reduction of activity throughout the brain has the potential to lower any aberrantly high thresholds for associative plasticity at synapses throughout the brain, and thereby reverse any learning deficits associated with such aberrantly high plasticity thresholds. This approach might even be useful in cases where the neural circuitry supporting a given behavior is not well characterized and the specific synapses responsible for the learning deficit are unknown. On lines 323-327 we compare this generalizable approach with the challenges of designing task- and circuit-specific approaches to reset the threshold for plasticity, particularly in circuits that are less well characterized than the oculomotor circuit.

It was not mentioned if L7-Fmr1 KO mice have behavior impairments that worsen with age or if Purkinje cells and the cerebellar microcircuit are intact throughout the lifespan.

At the adult ages used in our study (8-22 weeks), the oculomotor circuitry, including the Fmr1-deficient Purkinje cells, appears to be functionally intact because all of the oculomotor performance and learning tasks we tested were either normal, or could be restored to normal with brief behavioral and/or pharmacological pre-treatment.

Any degeneration of the Fmr1-deficient Purkinje cells or cerebellar microcircuit or additional behavioral impairments at older ages, if they should exist, would not alter our interpretation of the results from 8-22 week old adults regarding history- and activity-dependent changes in the capacity for LTD-dependent learning. Therefore, we leave the question of changes throughout the lifespan to investigators with an interest and expertise in development and/or aging.

Only a small handful of the scores of previous studies of the Fmr1 KO mouse model have investigated age-dependent effects; the reviewer may be interested in papers such as Tang et al., 2015 (doi: 10.1073/pnas.1502258112) or Martin et al., 2016 (doi: 10.1093/cercor/bhv031).

Connections between Purkinje cells and interneurons could also influence the behavior results found.

This comment is repeated below in a more general form (Reviewer 1, second to last comment)—please see our response there and lines 270-309 of the revised manuscript for a discussion of how concerns about “off-target” effects are mitigated by the high degree of specificity of the learning deficits and effects of pre-training for the specific learning tasks in which LTD has been previously implicated, and the very similar findings in two different lines of mice with enhanced LTD.

While males and females were both used for the current study, only 7 of each sex were analyzed, which could be underpowered. While it might be justified to combine sexes for this particular study, it would be worth understanding this model in more detail.

We performed additional analyses to address the question of whether there might be sex differences that were not detected because of the sample size.

(1) In a new figure, Fig. 1-figure supplement 1, we break out the results for male and female mice in separate plots, and show that all of the effects of both the KO of Fmr1 from the Purkinje cells and of pretreatment with diazepam that are observed in the full cohort are also statistically significant in just the subset of male mice, and just the subset of female mice (see Fig. 1-figure supplement 1 legend for statistics). In other words, qualitatively, there are no sex differences, and all of the conclusions of our manuscript are statistically valid in both male and female mice. This strengthens the justification for combining sexes for the specific scientific purposes of our study.

(2) We performed a power analysis to determine how many mice would be needed to determine whether the very, very small quantitative differences between male and female mice are significant. The analysis indicates that this would require upwards of 70 mice of each sex for WT mice (Cohen’s d, 0.6162; power 0.95) and upwards of 2500 mice of each sex for L7-Fmr1 KO mice (Cohen’s d, 0.0989; power 0.95). Since the very small quantitative sex differences observed in our cohorts would not alter our scientific conclusions or the possibility for clinical application to patients of both sexes, even if the small quantitative differences turned out to be significant, the very large number of animals needed did not seem warranted for the current scientific purposes. Researchers focused on sex differences may find a motivation to pursue this issue further.

Training was only shown up to 30 minutes and learning did not seem to plateau in most cases. What would happen if training continued beyond the 30 minutes? Would L7-Fmr1 KO mice catch-up to WT littermates? Nguyen-Vu

(1) For VOR learning, we used a 30 min training time because in our past (e.g., Boyden et al., 2003; Kimpo and Raymond, 2007; Nguyen-Vu et al., 2013; Nguyen-Vu et al., 2017) and current results, we find that VOR learning does plateau quite rapidly, with little or no additional adaptive change in the VOR observed between the tests of learning after 30 min vs 20 min of VOR-increase training, in WT or L7Fmr1 KO mice (Fig. 1A; WT, p=0.917; L7-Fmr1 KO, p=0.861; 20 vs. 30 min; Tukey). In the L7-Fmr1 KO mice, there is no significant high-frequency VOR-increase learning after 30 min training, and the mean VOR gain is even slightly lower on average (not significant) than before training (Fig. 1A, red). Therefore, we have no reason to expect that the L7-Fmr1 KO mice would catch up to WT after additional VOR-increase training.

(2) We have added new data on OKR adaptation, induced with 60 min of training (Fig. 5). The L7-Fmr1 KO mice exhibited impaired OKR adaptation, even with 60 min of training (p = 1.27x10-4, Tukey). In our experience, restraint for longer than 60 min produces a behavioral state that is not conducive to learning, as also reported by (Katoh and Yamagiwa, 2018), therefore longer training times were not attempted.

The pathway discussed as the main focus for VOR in this learning paradigm was connections between parallel fibers (PF) and Purkinje cells, but the possibility of other local or downstream circuitry being involved was not discussed. PF-Purkinje cell circuits were not directly analyzed, which makes this claim difficult to assess.

In the revised manuscript (lines 299-309), we have expanded our discussion of the possibility that loss of expression of Fmr1 from Purkinje cells in the Purkinje cell-specific L7-Fmr1 KO mice might influence other synapses or intrinsic properties of the Purkinje cells (including synapses from interneurons, as raised in this reviewer’s comment above), in addition to enhancing associative LTD at the parallel fiberPurkinje cell synapses.

It is a very general limitation of all perturbation studies, even cell-type specific perturbation studies as in the current case, that it is never possible to completely rule out “off-target” effects of the manipulation. Because of this, causality cannot be definitively concluded from correlations (e.g., between the effects of a perturbation observed at the cellular and behavioral level), and therefore we make no such claim in our manuscript. Rather, we conclude that our results “provide evidence for,” “support,” “predict,” or “are consistent with” the hypothesis of a history- and activity-dependent change in the threshold for associative LTD at the parallel fiber-Purkinje cells.

That said, perturbation is still one of the major tools in the experimental toolbox, and there are approaches for mitigating concern about off-target effects. We highlight three aspects of our experimental design that accomplish this (lines 184-228, 256-309). First, we show nearly identical learning impairments and effects of behavioral pretreatment in lines of mice with two completely different molecular manipulations that have the common effect of enhancing PF-Purkinje cell LTD, but are likely to have different off-target cellular effects on the Purkinje cells and their synapses. Second, we show that the learning impairments were highly specific to oculomotor learning tasks in which PF-Purkinje cell LTD was previously implicated, with no such effects on three other oculomotor learning tasks that depend on the same region of the cerebellum and oculomotor circuitry. In the original submission, we provided data for one LTDdependent oculomotor learning task, high-frequency VOR-increase learning; in the revised manuscript we provide new data for a second LTD-dependent oculomotor learning task, optokinetic reflex adaptation, with nearly identical results (Fig. 5). Third, we show that the effects of diazepam pre-treatment were highly specific to the same two LTD-dependent oculomotor learning tasks and also highly specific to the L7-Fmr1 KO mice with enhanced LTD and not WT mice. These three features of the experimental design are not common in studies of learning, especially in combination. On lines 256-309, we provide an expanded discussion of how together, these three features of the design strengthen the evidence that the learning impairments and effects of diazepam pre-treatment on learning are related to LTD at the PF-Pk synapses, while acknowledging the possibility of other effects on the circuit.

The authors mostly achieved their aim and the results support their conclusion and proposed hypothesis. This work will be impactful on the field as it uses a new Purkinje-cell specific mouse model to study a classic cerebellar task. The use of diazepam could be further analyzed in other genetic models of neurodevelopmental disorders to understand if effects on LTD can rescue other pathways and behavior outcomes.

We agree that the present findings are potentially relevant for a very wide array of behavioral tasks, disease models, and brain areas beyond the specific ones in our study, and we make this point on lines 310-338 of the revised manuscript.

**Reviewer #2 (Public Review):**
This manuscript explores the seemingly paradoxical observation that enhanced synaptic plasticity impairs (rather than enhances) certain forms of learning and memory. The central hypothesis is that such impairments arise due to saturation of synaptic plasticity, such that the synaptic plasticity required for learning can no longer be induced. A prior study provided evidence for this hypothesis using transgenic mice that lack major histocompatibility class 1 molecules and show enhanced long-term depression (LTD) at synapses between granule cells and Purkinje cells of the cerebellum. The study found that a form of LTD-dependent motor learning-increasing the gain of the vestibulo-ocular reflex (VOR)-is impaired in these mice and can be rescued by manipulations designed to "unsaturate" LTD. The present study extends this line of investigation to another transgenic mouse line with enhanced LTD, namely, mice with the Fragile X gene knocked out. The main findings are that VOR gain increased learning is selectively impaired in these mice but can be rescued by specific manipulations of visuomotor experience known to reverse cerebellar LTD. Additionally, the authors show that a transient global enhancement of neuronal inhibition also selectively rescues gain increases learning. This latter finding has potential clinical relevance since the drug used to boost inhibition, diazepam, is FDA-approved and commonly used in the clinic. The evidence provided for the saturation is somewhat indirect because directly measuring synaptic strength in vivo is technically difficult. Nevertheless, the experimental results are solid. In particular, the specificity of the effects to forms of plasticity previously shown to require LTD is remarkable. The authors should consider including a brief discussion of some of the important untested assumptions of the saturation hypothesis, including the requirement that cerebellar LTD depends not only on pre- and postsynaptic activity (as is typically assumed) but also on the prior history of synaptic activation.

We thank the reviewer for this exceptionally clear and concise assessment of the findings and strengths of the manuscript.

We agree that one of the most “remarkable” aspects of our findings is the specificity of the effects for oculomotor learning tasks for which there is the strongest previous evidence for a role of PF-Purkinje cell LTD. In the original manuscript, we tested just one LTD-dependent oculomotor learning task, highfrequency VOR increase learning; in the revised manuscript, we strengthen the case for LTD-dependent task specificity by adding new data (Fig. 5) showing the same effects for OKR adaptation, an additional LTD-dependent oculomotor learning task.

The reviewer’s suggestion to include discussion of “untested assumptions”, “including the requirement that cerebellar LTD depends not only on pre- and postsynaptic activity (as is typically assumed) but also on the prior history of synaptic activation” prompted us to more deeply consider the broader implications of our results, and extensively revise the Discussion accordingly. We clarify that we consider historydependent changes in the threshold for LTD to be a prediction of the behavioral and pharmacological findings (lines 339-347, 356) rather than an assumption. In addition, we highlight the broader implications of the results by putting them in the context of work in other brain areas on historydependent changes in the threshold for plasticity, i.e., metaplasticity, going back to the seminal Bienenstock-Cooper-Munro (BCM; year) theory (lines 348-378).

**Reviewer #1 (Recommendations for The Authors):**
The text and figures are very clear to read, but there are a couple of questions that remain:The concentrations chosen for diazepam are not well described and it is unclear why the concentrations jump from 2.5 mg/kg to 0.5 mg/kg. Please add an explanation for these concentrations and if any additional behavior outcomes were observed.

Our choice of diazepam concentrations was guided by the concentrations reported in the literature to be effective in mice, which suggest that a higher dose (2 mg/kg) can have additional effects not observed with a lower effective dose (0.5 mg/kg) (Pádua-Reis et al, 2021). Since we did not know how much enhancement of inhibition/suppression of activity might be necessary to substantially reduce the induction of PF-Purkinje cell LTD, we did pilot experiments to test concentrations at the low and high ends of the doses typically used in mice. These pilot experiments revealed that a lower dose of 0.4 or 0.5 mg/kg was comparable to the higher dose of 2.5 mg/kg in suppressing VOR-increase learning 2 hours after administration (Fig. 3 – figure supplement 2). Anecdotally, we observed higher levels of locomotor activity and other abnormal cage behavior during the period immediately after administration of the higher compared to the lower dose. To limit these side effects and any possibility of dependence, we used only the lower dose in all subsequent experiments. We clarify this rationale for using a lower dose in the legend of Fig. 3 – figure supplement 2.

Figure 4 describes low-frequency VOR, but the paragraph discussing these results (line 191) mentions high-frequency VOR-increase learning. It is unclear where the results are for the high-frequency data. Please include or rephrase for clearer understanding.

In the revised manuscript, we clarify that the 1 Hz vestibular and visual stimuli used in Figs. 1-3 is the

“high” frequency, which yields different results than the “low” frequency of 0.5 Hz (Fig. 4), as also observed in Boyden et al 2006, and Nguyen-Vu et al, 2017.

**Reviewer #2 (Recommendations For The Authors):**
The authors should consider including a brief discussion of some of the important untested assumptions of the saturation hypothesis, including the requirement that cerebellar LTD depends not only on pre- and postsynaptic activity (as is typically assumed) but also on the prior history of synaptic activation.

We thank the reviewer for this comment, which, along with your public comments, inspired us to thoroughly reconsider and revise our Discussion. We think this has greatly improved the manuscript, and will substantially increase its appeal to a broad segment of the neuroscience research community, including computational neuroscientists as well as those interested in synaptic physiology, learning and memory, or plasticity-related brain disorders including autism.

Note that we consider the idea that ”LTD depends not only on pre- and post- synaptic activity but also on the prior history of synaptic activation” to be the central prediction of the threshold metaplasticity hypothesis rather than an assumption, and in the revised manuscript we explicitly refer to this as a prediction (line 339, 356). We also added a discussion of multiple known cellular phenomena in the Purkinje cells and their synapses that can regulate LTD and thus represent candidate mechanisms for LTD threshold metaplasticity (lines 339-347). Again, sincere thanks for prompting us to write a vastly improved Discussion section.

**Editor's note**:Should you choose to revise your manuscript, please include full statistical reporting including exact pvalues wherever possible alongside the summary statistics (test statistic and df) and 95% confidence intervals. These should be reported in the main text for all key questions and not only when the p-value is less than 0.05.

We have added exact p-values throughout the manuscript.

References

Albergaria C, Silva NT, Pritchett DL, Carey MR. (2018). Locomotor activity modulates associative learning in mouse cerebellum. Nat Neurosci.21:725-735. doi: 10.1038/s41593-018-0129-x.

Abraham WC, Mason-Parker SE, Bear MF, Tate WT. (2001). Heterosynaptic metaplasticity in the hippocampus in vivo: A BCM-like modifiable threshold for LTP. Proc Natl Acad Sci USA. 98:1092410929.

Bienenstock E, Cooper L, Munro P. (1982). Theory for the development of neuron selectivity: orientation specificity and binocular interaction in visual cortex. J Neurosci. 2:32-48. https://doi.org/10.1523/JNEUROSCI.02-01-00032.1982

Brett J, Murnion B. (2015). Management of benzodiazepine misuse and dependence. Aust Prescr.38:152155. doi: 10.18773/austprescr.055.

Boyden ES, Raymond JL. (2003). Active Reversal of Motor Memories Reveals Rules Governing Memory Encoding. Neuron.39:1031-1042. https://doi.org/10.1016/S0896-6273(03)00562-2

Boyden ES, Katoh A, Pyle JL, Chatila TA, Tsien RW, Raymond JL. (2006). Selective engagement of plasticity mechanisms for motor memory storage. Neuron. 51:823-834. https://doi.org/10.1016/j.neuron.2006.08.026

Desai NS, Cudmore RH, Nelson SB, Turrigiano GG. (2002). Critical periods for experience-dependent synaptic scaling in visual cortex. Nat Neurosci. 5:783-789. doi: 10.1038/nn878.

Fong M, Duffy KR, Leet MP, Candler CT, Bear MF. (2021). Correction of amblyopia in cats and mice after the critical period. ELife.10:e70023. https://doi.org/10.7554/eLife.70023

Hamada M, Terao Y, Hanajima R, Shirota Y, Nakatani-Enomoto S, Furubayashi T, Matsumoto H, Ugawa Y. (2008). Bidirectional long-term motor cortical plasticity and metaplasticity induced by quadripulse transcranial magnetic stimulation. J Physiol. 586:3927-3947. doi: 10.1113/jphysiol.2008.152793.

Katoh A, Yamagiwa A. (2018). Inhibition of PVN neurons influences stress-induced changes of motor learning in the VOR. Society for Neuroscience. Online Program No. 067.14.

Kimpo RR, Raymond JL. (2007). Impaired motor learning in the vestibulo-ocular reflex in mice with multiple climbing fiber input to cerebellar Purkinje cells. J Neurosci. 27:5672-5682. doi:

10.1523/JNEUROSCI.0801-07.2007.

Kirkwood A, Rioult MG, Bear MF. (1996). Experience-dependent modification of synaptic plasticity in visual cortex. Nature. 381:526–528. https://doi.org/10.1038/381526a0

Koekkoek SK, Yamaguchi K, Milojkovic BA, Dortland BR, Ruigrok TJ, Maex R, De Graaf W, Smit AE, VanderWerf F, Bakker CE, Willemsen R, Ikeda T, Kakizawa S, Onodera K, Nelson DL, Mientjes E, Joosten M, De Schutter E, Oostra BA, Ito M, De Zeeuw CI. (2005). Deletion of FMR1 in Purkinje Cells Enhances Parallel Fiber LTD, Enlarges Spines, and Attenuates Cerebellar Eyelid Conditioning in Fragile X Syndrome. Neuron. 47:339–352. https://doi.org/10.1016/j.neuron.2005.07.005

Le Friec A, Salabert AS, Davoust C, Demain B, Vieu C, Vaysse L, Payoux P, Loubinoux I. (2017). Enhancing Plasticity of the Central Nervous System: Drugs, Stem Cell Therapy, and Neuro-Implants. Neural Plast. 2017:2545736. doi: 10.1155/2017/2545736.

Leet MP, Bear MF, Gaier ED. (2022). Metaplasticity: a key to visual recovery from amblyopia in adulthood? Curr Opin Ophthalmol. 33:512–518. https://doi.org/10.1097/ICU.0000000000000901

Martin HGS, Lassalle O, Brown JT, Manzoni OJ. (2016). Age-Dependent Long-Term Potentiation Deficits in the Prefrontal Cortex of the Fmr1 Knockout Mouse Model of Fragile X Syndrome. Cereb Cortex. 26:2084–2092. doi: 10.1093/cercor/bhv031.

Montgomery JM, Madison DV. (2002). State-dependent heterogeneity in synaptic depression between pyramidal cell pairs. Neuron. 33:765-777. doi: 10.1016/s0896-6273(02)00606-2.

Nguyen-Vu TDB, Kimpo RR, Rinaldi JM, Kohli A, Zeng H, Deisseroth K, Raymond JL. (2013). Cerebellar Purkinje cell activity drives motor learning. Nat Neurosci. 16:1734-1736. doi:10.1038/nn.3576.

Nguyen-Vu TB, Zhao GQ, Lahiri S, Kimpo RR, Lee H, Ganguli S, Shatz CJ, Raymond JL. (2017). A saturation hypothesis to explain both enhanced and impaired learning with enhanced plasticity. ELife. 6:e20147. https://doi.org/10.7554/eLife.20147

Pádua-Reis M, Nôga DA, Tort ABL, Blunder M. (2021). Diazepam causes sedative rather than anxiolytic effects in C57BL/6J mice. Sci Rep. 2021;11:9335.

Redondo RL, Morris RG. (2011). Making memories last: the synaptic tagging and capture hypothesis. Nat Rev Neurosci. 2011 Jan;12(1):17-30. doi: 10.1038/nrn2963.

Singh A, Nagpal R, Mittal SK, Bahuguna C, Kumar P. (2017). Pharmacological therapy for amblyopia. Taiwan J Ophthalmol. 7:62-69. doi: 10.4103/tjo.tjo_8_17.

Tang B, Wang T, Wan H, Han L, Qin X, Zhang Y, Wang J, Yu C, Berton F, Francesconi W, Yates JR 3rd, Vanderklish PW, Liao L. (2015). Fmr1 deficiency promotes age-dependent alterations in the cortical synaptic proteome. Proc Natl Acad Sci USA. 112:E4697-E4706. doi: 10.1073/pnas.1502258112.

Van Rossum MC, Bi GQ, Turrigiano GG. (2000). Stable Hebbian learning from spike timing-dependent plasticity. J Neurosci. 2000 Dec 1;20(23):8812-21. doi: 10.1523/JNEUROSCI.20-23-08812.2000.

Wang W, Nakadate K, Masugi-Tokita M, Shutoh F, Aziz W, Tarusawa E, Lorincz A, Molnár E, Kesaf S, Li YQ, Fukazawa Y, Nagao S, Shigemoto R. (2014). Distinct cerebellar engrams in short-term and long-term motor learning. Proc Natl Acad Sci U S A. 2014 Jan 7;111(1):E188-93. doi: 10.1073/pnas.1315541111.

Yamaguchi T, Moriya K, Tanabe S, Kondo K, Otaka Y, Tanaka S. (2020). Transcranial direct-current stimulation combined with attention increases cortical excitability and improves motor learning in healthy volunteers. J Neuroeng Rehabil. 17:23. doi: 10.1186/s12984-020-00665-7.